# Diffusion-Based Adversarial Sample Generation for Improved Stealthiness and Controllability

**Haotian Xue** [*1]     **Alexandre Araujo** [2]     **Bin Hu** [3]     **Yongxin Chen** [1]

[1] Georgia Institute of Technology
[2] New York University
[3] University of Illinois Urbana-Champaign

## Abstract

Neural networks are known to be susceptible to adversarial samples: small variations of natural examples crafted to deliberately mislead the models. While they can be easily generated using gradient-based techniques in digital and physical scenarios, they often differ greatly from the actual data distribution of natural images, resulting in a trade-off between strength and stealthiness. In this paper, we propose a novel framework dubbed Diffusion-Based Projected Gradient Descent (Diff-PGD) for generating realistic adversarial samples. By exploiting a gradient guided by a diffusion model, Diff-PGD ensures that adversarial samples remain close to the original data distribution while maintaining their effectiveness. Moreover, our framework can be easily customized for specific tasks such as digital attacks, physical-world attacks, and style-based attacks. Compared with existing methods for generating natural-style adversarial samples, our framework enables the separation of optimizing adversarial loss from other surrogate losses (*e.g.*, content/smoothness/style loss), making it more stable and controllable. Finally, we demonstrate that the samples generated using Diff-PGD have better transferability and anti-purification power than traditional gradient-based methods. Code is available at https://github.com/xavihart/Diff-PGD

## 1 Introduction

Neural networks have demonstrated remarkable capabilities in learning features and achieving outstanding performance in various downstream tasks. However, these networks are known to be vulnerable to subtle perturbations, called adversarial samples (adv-samples), leading to important security concerns for critical-decision systems [16]. Numerous threat models have been designed to generate adversarial samples for both digital space attacks [31, 5], physical world attacks [26, 14], and also for generating more customized adv-samples given prompts like attack region and style reference [12, 17]. However, most of these models have not been designed to maintain the realism of output samples, resulting in adversarial samples that deviate significantly from the distribution of natural images [22]. Indeed, in the setting of digital world attacks, such as those involving RGB images, higher success rates, and transferability are associated with larger changes in the generated samples, leading to a stealthiness-effectiveness trade-off. Additionally, perturbations for physical-world attacks are substantial, non-realistic, and easily noticeable by humans [47].

Recent works, for both digital and physical settings, have proposed methods to generate more realistic adversarial samples. For example, [29, 24] introduced a method to optimize the perturbations added to clean images in semantic space, [23, 26] proposed a GAN-based approach or other prior knowledge [33] to increase the realism of generated samples. Although the GAN-based approach

---

* Correspondence to: `htxue.ai@gatech.edu`

37th Conference on Neural Information Processing Systems (NeurIPS 2023).

Table 1: **Properties of Diff-PGD vs Other Attacks**: We summarize six metrics for adv-sample generation. "Stealthiness" measures whether the adversarial perturbations can be detected by human observers. "Scenarios" describes the setting in which the method can be applied: $D$ corresponds to digital attacks and $P$ for physical attacks. "Controllability" measures whether the method can support customized prompts *e.g.*, mask/style reference. "Anti-Purify" measures the ability of the samples to avoid being purified. "Transferability" measures the generalization of the attack to different models. Finally, "Stability" describes the stability of the optimization. (-) stands for non-consideration.

| Methods | Stealthiness | Scenarios | Controllability | Anti-Purify | Transferability | Stability |
|---|---|---|---|---|---|---|
| PGD [31] | ** | D | * | * | * | ** |
| AdvPatch [4] | * | P | ** | (-) | (-) | ** |
| NatPatch [23] | ** | P | * | (-) | (-) | ** |
| AdvArt [17] | * | P | ** | (-) | (-) | ** |
| AdvCam [12] | ** | D/P | ** | (-) | (-) | * |
| **Diff-PGD (Ours)** | ** | D/P | ** | ** | ** | ** |

can generate realistic images, the adversarial examples are sampled from noise and therefore lack controllability. A subsequent line of research on realistic adversarial samples [22] has introduced the concept of semantic adversarial samples. These are unbounded perturbations that deceive the model, while ensuring the modified image has the same semantic information as the original image. While transformations in semantic space can reduce the high-frequency patterns in pixel space, this approach still suffers from color jittering or image distortion which results in lower stealthiness [12]. Further, this approach needs careful tuning of the hyperparameters making the training process unstable.

In this paper, we propose a novel framework, which uses an off-the-shelf diffusion model to guide the optimization of perturbations, thus enabling the generation of adversarial samples with higher stealthiness. The core design of our framework is Diffusion-Based Projected Gradient Descent (Diff-PGD), which shares a similar structure with PGD but changes the input of the target classifier $f_\theta$ to be purified version $x_0$ of the original input $x$. To the best of our knowledge, we are the first to use the Diffusion Model to power the generation of adv-samples. Our framework can also be easily adapted to some customized attacks given masks or style references, thereby enhancing its controllability. Our framework (Figure 1(c)) separates the customized process with prompt $p$ (region mask, style, etc.) with adversarial sample generation, overcoming the drawbacks of previous pipelines: the traditional gradient-based methods (Figure 1(a)) cannot guarantee to generate naturalistic samples when the perturbation level is large; the joint optimization with multiple losses like adv-loss $l_{adv}$, style loss $l_z$ and realism loss $l_r$ (see Figure 1(b)) is unstable to train and still tend to generate perceptible artifacts. Through extensive experiments in scenarios like digital attacks, masked region attacks, style-guided attacks, and physical world attacks, we show that our proposed framework can effectively generate realistic adv-samples with higher stealthiness and controllability. We further show that the Diff-PGD can help generate adversarial samples with higher transferability than traditional methods without gradient restriction. Our contribution can be summarized as follows:

1. We propose a novel framework Diff-PGD, combing the strong prior knowledge of the diffusion model into adv-sample generation, which helps generate adv-samples with high stealthiness and controllability. Adv-samples generated by Diff-PGD have good properties as described in Table 1.

2. We show that Diff-PGD can be effectively applied to many tasks including digital attacks, customized attacks, and physical-world attacks, outperforming baseline methods such as PGD, AdvPatch, and AdvCam.

3. We explore the transferability and anti-purification properties of Diff-PGD samples and show through an experimental evaluation that adversarial samples generated by Diff-PGD outperform the original PGD approach.

## 2   Related Work

Adversarial attacks aim at maximizing the classification error of a target model without changing the semantics of the images. In this section, we review the types of attacks and existing works.

**Norm bounded attacks.** In the digital space, one can easily generate adversarial samples by using gradient-based methods such as Fast Gradient Sign Method (FGSM) [16] or Projected Gradient Decent (PGD) [31]. These methods aim at maximizing the loss of the target model with respect to the input and then projecting the perturbation to a specific $\ell_p$ ball. It has been observed that these adversarial samples tend to diverge from the distribution of natural images [50]. Based on this observation, [34] have shown that these attacks can be "purified" using pre-trained diffusion models.

**Semantic attacks.** Some recent works operate semantic transformation to an image: [22] generate adv-samples in the HSV color space, [48, 13] generate adv-samples by rotating the 2D image or changing its brightness, and [52, 29, 10] try to generate semantic space transformations with an additional differentiable renderer. All these methods are far from effective; they either need additional modules or have color jitters and distortion [12].

**Customized Attacks with Natural-Style.** Significant efforts have been made toward generating customized adversarial samples (given region and/or style reference) with natural-style: AdvCAM [12] optimize adversarial loss together with other surrogate losses such as content loss, style loss, and smoothness loss to make the output adversarial sample to be realistic, [23, 26, 55] sample from latent space and then use a Generative Adversarial Network (GAN) to guarantee the natural style of the output sample to fool an object detector, and AdvArt [17] customized the generation of the adversarial patch with a given art style. However, all these methods share some common issues: optimizing adversarial loss and other surrogate losses (for content-preserving or customized style) needs careful balance and still results in unnatural artifacts in the output samples. Our methods use the diffusion model as a strong prior to better ensure the realism of generated samples.

**Attacks in physical-world.** For physical-world attacks, minor disturbances in digital space often prove ineffective due to physical transformations such as rotation, lighting, and viewing distance. In this context, adding adversarial patches to the scene is a common technique to get the target model to misclassify [4, 14, 1, 25]. Physical-world attacks do not impose any constraints either on the amount of perturbation or on the output style, thus, they tend to produce highly unnatural images that can be easily detected by human observers (*i.e.*, lack stealthiness).

## 3 Background

**Diffusion Model.** Diffusion Models (DMs) [19, 41] have demonstrated superior performance in many tasks such as text-to-image generation [37, 2, 38], video/story generation [21, 20, 35], 3D generation [28, 36] and neural science [44, 42]. Pre-trained DMs provide valuable prior knowledge that can be exploited for adversarial robustness. In this context, several works have proposed defense mechanisms such as adversarial purification [34, 53, 43, 34, 27] or DM-enhanced certified robustness [6, 46, 45].

The Denoised Diffusion Probabilistic Model (DDPM) [19] is a discretized version of DMs and works as follows. Suppose $x_0 \sim p(x_0)$ is a sample from the distribution of natural images. The forward diffusion process gradually adds Gaussian noise to $x_0$, generating noisy samples $[x_1, x_2, \cdots, x_t, \cdots, x_T]$ in $T$ steps, following a Markov process defined as $q_M(x_t \mid x_{t-1}) = \mathcal{N}(x_t; \sqrt{1 - \beta_t}\, x_{t-1}, \beta_t \mathbf{I})$ where $\mathcal{N}$ denotes Gaussian distribution $\beta_t$ are fixed values growing from 0 to 1. By accumulating single step $q_M$ we have $q(x_t \mid x_0)$ as

$$q(x_t \mid x_0) = \mathcal{N}(x_t; \sqrt{\bar{\alpha}_t}\, x_{t-1}, (1 - \bar{\alpha}_t)\mathbf{I}) \tag{1}$$

where $\alpha_t = 1 - \beta_t$ and $\bar{\alpha}_t = \Pi_{s=1}^t \alpha_s$. When $\bar{\alpha}_t \approx 0$, the distribution of $x_T$ becomes an isotropic Gaussian.

The reverse process aims at generating samples from the target data distribution from Gaussian noise $x_T \sim \mathcal{N}(0, \mathbf{I})$ using the reversed diffusion process. The reverse model $p_\phi(x_{t-1} \mid x_t)$ can be trained by optimizing the usual variational bound on negative log likelihood. Using the re-sampling strategy proposed in [19], we can simplify the optimization of $\phi$ into a denoising process, by training a modified U-Net as a denoiser. Then we can generate $x_0$ with high fidelity by sampling on the reversed diffusion:

$$p(x_{0:T}) = p(x_T) \prod_{t=1}^T p_\phi(x_{t-1} \mid x_t). \tag{2}$$

For the reverse process, we define $R_\phi$ parameterized by $\phi$ as the function to denoise $x_t$ and then get next sampled value as $x_{t-1} = R_\phi(x_t, t)$.

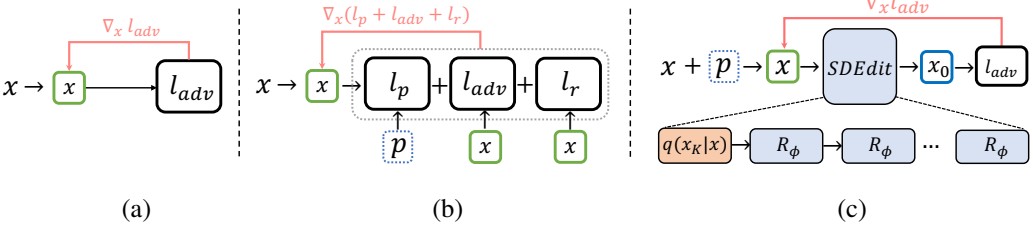

Figure 1: **Comparison of Different Pipelines**: (a) Traditional gradient-based adversarial sample generation, $x$ is the sample to be optimized, $l_{adv}$ is adversarial loss. (b) Customized adversarial sample generation with natural style (determined by prompt $p$): joint optimization of adversarial loss with other surrogate losses like prompt loss $l_p$ (e.g. style loss) and realistic loss $l_r$ (e.g. content loss, smooth loss). (c) Our proposed diffusion-based framework, $q$ is forward diffusion and $R_\phi$ is backward denoising, $x_0$ is the denoised sample.

**SDEdit.**    The key idea of Stochastic Differential Editing (SDEdit) [32] is to diffuse the original distribution with $x_K \sim q(x_K \mid x)$ and then run the reverse denoising process $R_\phi$ parametrized by $\phi$ in a Diffusion Model to get an edited sample:

$$\text{SDEdit}(x, K) = R_\phi(\dots R_\phi(R_\phi(x_K, K), K-1)\dots, 0). \qquad (3)$$

Given a raw input, SDEdit first runs $K$ steps forward and then runs $K$ steps backward. This process works as a bridge between the input distribution (which always deviates from $p(x_0)$) and the realistic data distribution $p(x_0)$, which can then be used in tasks such as stroke-based image synthesis and editing. It can also be used in adversarial samples purification [34, 53].

## 4    Diffusion-based Projected Gradient Descent

We present Diffusion-based Projected Gradient Descent (Diff-PGD), a novel method to generate adversarial samples that looks realistic, using a pre-trained diffusion model. First, we present basic Diff-PGD that works on global image inputs, which shares the same bound settings with $\ell_\infty$-PGD in Section 4.1. Then, in Section 4.2 we generalize Diff-PGD to scenarios where only regional attacks defined by a customized mask are possible, using re-painting skills in the diffusion model [30]. Finally, we present extensions of Diff-PGD to customized attacks guided by image style in Section 4.3 and physical world attack in Section 4.4.

In the rest of the paper, we use superscript $i$ or $t$ to denote *iteration of optimization*, and subscript $i$ to denote the *timestep of diffusion model*.

### 4.1    Diffusion-based Projected Gradient Descent

Given a clean image $x$ with label $y$ and a target classifier $f_\theta$ parameterized by $\theta$ to be attacked, our goal is to generate an adversarial sample $x_{adv}$, which fools the model. The traditional gradient-based methods use the gradient of adversarial loss $g = \nabla_x l(f_\theta(x), y)$, where $f_\theta(x)$ return the logits. We also denote the loss as $l_{adv}$ for simplicity to optimize $x$ by iterations. The $t$-step update in the PGD with stepsize $\eta$ and $n$ iterations reads

$$x^{t+1} = \mathcal{P}_{B_\infty(x, \epsilon)} \left[ x^t + \eta \, \text{sign} \nabla_{x^t} l(f_\theta(x^t), y) \right] \qquad (4)$$

where $\mathcal{P}_{B_\infty(x, \epsilon)}(\cdot)$ is the projection operator on the $\ell_\infty$ ball. Intuitively, the gradient will "guide" the sample $x^t$ away from the decision boundary and deviate from the real distribution $p(x_0)$ [29]. This out-of-distribution property makes them easier to be purified using some distribution-based purification methods [34], which considerably restricts the strength of the attacks.

Instead of using $l(f_\theta(x), y)$ as the optimization cost, we turn to use logits of purified image $f_\theta(x_0)$ as our optimization target. Here $x_0$ can be obtained using SDEdit with $K$ reverse steps. The update step becomes

$$x_0^t = \text{SDEdit}(x^t, K) \quad \text{and} \quad x^{t+1} = \mathcal{P}_{B_\infty(x, \epsilon)} \left[ x^t + \eta \, \text{sign} \nabla_{x^t} l(f_\theta(x_0^t), y) \right]. \qquad (5)$$

The parameter $K$ can be used to control the edition-content preference [32]: a large $K$ value brings more editions and a small $K$ value preserves more original content, and empirically,

---

**Algorithm 1** Diff-rPGD

---

**Require:** Target classifier $f_\theta$, original image $x$, mask $M$, denoiser $D_\phi$, # of reverse SDEdit steps $K$, iterations $n$, stepsize $\eta$, clip value $\epsilon$ (when $M = 1$ it reduces to Diff-PGD)

$x^0 = x_0^0 = x$, $x_c = x$

**for** $t = 0, 1, 2, \ldots, n-1$ **do**

 $x_K^t \sim q(x_K^t | x^t)$          ▷ Sample $x_K^t$ from $q(x_K^t | x^t)$ in each PGD iteration

 **for** $i = K-1, \ldots, 0$ **do**

  $x_i^t = R_\phi(x_{i+1}^t)$        ▷ Apply denoiser $R_\phi$ to $x_{i+1}^t$ in each SDEdit iteration

  $x_i^t \sim M \circ x_i^t + (1 - M) \circ q(x_i^t | x^t)$ ▷ Sample from masked combination of $x_i^t$ and $q(x_i^t | x^t)$

 **end for**

 $g = \nabla_{x^t} l[f_\theta(M \circ x_0^t + (1 - M) \circ x_c)]$       ▷ Compute the gradient

 $x^{t+1} = \Pi_{x,\epsilon}[x^t + \eta \, M \circ \text{sign}(g)]$          ▷ PGD update

**end for**

$x_0^n = \text{rSDEdit}(x^n)$          ▷ Apply reverse SDEdit to the final $x^n$

**return** $x^n, x_0^n$       ▷ Return the final adversarial example and the denoised version

---

$K = 0.1 \sim 0.3T$ works well. Since we need to compute a new adversarial gradient $g_{\text{diff}} = \nabla_x R_\phi(\ldots R_\phi(R_\phi(x_t, K), K-1) \ldots, 0)$ through back-propagation, $K$ cannot be too large due to GPU memory constraint. Thus we turn to adopt some speed-up strategies like DDIM [40] to first sub-sample original $T$ into $T_s$, and then use scaled $K_s \ll K$ to do SDEdit (*e.g.*, $T = 1000, K = 100, T_s = 40, K_s = 4$). Other fast sampling schemes [54] for DMs may accelerate our algorithms further.

After $n$ iterations, Diff-PGD generates two outputs ($x_0^n$ and $x^n$) that are slightly different from each other: 1) $x_0^n$, according to the definition of the loss function, is realistic and adversarial. 2) $x^n$ is an optimized perturbed image to which the adversarial gradient is directly applied. It follows the PGD $\ell_\infty$ bound, and according to the loss function Equation (5), hard to be purified by SDEdit.

### 4.2 Extended Diff-PGD into Region Attack

In some scenarios, we need to maintain parts of the image unchanged and only generate adversarial perturbations in defined regions. This is used in customized attacks, *e.g.*, attack the masked region with a given natural style. Here we consider the sub-problem to attack the masked region (determined by mask $M$) of a given image with PGD, noted as region-PGD (rPGD). Specifically, the output adv-samples should satisfy: $(1 - M) \circ x_{adv} = (1 - M) \circ x$. Obviously, when $M$ is valued all ones, the rPGD reduces to the original PGD.

For the previous gradient-based method, it is easy to modify $\nabla_x f_\theta(x)$ into $\nabla_x f_\theta(M \circ x + (1-M) \circ x_c)$, where $x_c = x$ is a copy of the original image with no gradient flowing by. It is not surprising the perturbed regions will show a large distribution shift from other regions, making the figure unnatural.

For Diff-PGD, instead of using the original SDEdit, we modify the reverse process using a replacement strategy [30, 41] to get a more generalized method: Diff-rPGD. Similarly, when $M$ has valued all ones, it reduces to Diff-PGD. The pseudo-code for Diff-rPGD is shown in Alg. 1, where the replacement strategy can give a better estimation of intermediate samples during the reverse diffusion. This strategy can make the patches in the given region better align with the remaining part of the image in the reversed diffusion process.

### 4.3 Customized Adversarial Sample Generation

As previously mentioned, Diff-PGD allows for the optimization of the adversarial loss while simultaneously preserving reality. This property helps us avoid optimizing multiple losses related to realism at the same time, as is used in previous methods [12, 17, 26]. Naturally, this property benefits the generation of highly customized adv-samples with prompts (e.g., mask, style reference, text), since we can separate the generation into two steps: 1) generate samples close to given prompts (from $x$ to $\hat{x}_s$) and 2) perturb the samples to be adversarial and at the same time preserving the authenticity (from $\hat{x}_s$ to $x_0^n$). Here we follow the task settings in natural-style adv-sample generation with style reference in [12]: given the original image $x$, mask $M$, and style reference $x_s$, we seek to generate adv-samples $x_{adv}$ which shows the similar style with $x_s$ and is adversarial to the target classifier $f_\theta$.

The style distance, as defined in [15], is composed of the differences between the Gram matrix of multiple features along the deep neural network, where $H_s$ are layers for feature extraction (*e.g.* Conv-layers), and $G$ is used to calculate the Gram matrix of the intermediate features. The style loss between $x$ and $x_s$ as $l_s(x, x_s)$ is

$$l_s(x, x_s) = \sum_{h \in H_s} \|G(f_h(x)) - G(f_h(x_s))\|_2^2 \tag{6}$$

AdvCAM [12] uses $l = \lambda_1 l_s + \lambda_2 l_{adv} + \lambda_3 l_r$ as the optimization loss where $l_r$ includes losses like smooth loss and content loss, all designed to make the outputs more realistic. It is tricky to balance all these losses, especially the adversarial loss and the style loss. We emphasize that it is actually not necessary to train the adversarial loss $l_{adv}$ together with $l_s$. This is due to the fact that Diff-PGD can guarantee the natural style of the output adversarial samples. Thus, the optimization process can be streamlined by first optimizing the style loss, and subsequently running Diff-PGD/Diff-rPGD. The high-level pipeline can be formulated as follows:

$$x + p \rightarrow \hat{x}_s \quad \text{and} \quad (x^n, x_0^n) = \text{Diff-rPGD}(\hat{x}_s, K, f_\theta) \tag{7}$$

The first equation serves as a first stage of optimization to acquire region styled $\hat{x}_s$ given prompts $p = (M, x_s)$. This stage can be implemented with other methods (*e.g.* generative-model-based image editing) and we do not need to worry about $f_\theta$ at this stage. Then we use Diff-rPGD to generate the adversarial sample $x_{adv}$ given $\hat{x}_s$ and other necessary inputs. Through experiments, we find that there are two major advantages of this pipeline: 1) the generation is much more stable 2) for some cases where the style loss is weak to guide the style transfer (generate samples that do not make sense), we can adjust $K$ so that the diffusion model can help to generate more reasonable samples.

### 4.4 Physical World Attack with Diff-PGD

We can also apply Diff-PGD to physical world attacks. First, for physical-world attacks, we need to introduce a *Physical Adapter* in order to make the attack more robust in physical scenarios. Second, instead of using $\ell_\infty$-PGD as a bounded attack (*e.g.* $\epsilon = 128/255$, which is proved to be ineffective in [12]), we turn to utilize the same idea of Diff-PGD, but apply it as Diff-Phys, a patch generator with guidance from a diffusion model. We define our new loss as:

$$l_{\text{Diff-Phys}}(x) = l_{adv}(\mathbb{E}_{\tau \sim \mathcal{T}}[\tau(\text{SDEdit}(x, K))]) \tag{8}$$

Here, the physics adapter is included by adopting a transformation set $\mathcal{T}$, which is utilized to account for variations in image backgrounds, random translations, changes in lighting conditions, and random scaling. We can apply Diff-PGD to patch attacks

$$x^* = \arg \min_x l_{\text{Diff-Phys}}(x) \quad \text{and} \quad x_0^* = \text{SDEdit}(x^*, K). \tag{9}$$

The pipeline works as follows. We first gather background data and define the scope of translations and lighting conditions, to get $\mathcal{T}$. Then we optimize our patch in simulation to get $x_0^*$. Finally, we print $x_0^*$ out, stick it to the target object, take photos of the object, and test it on the target classifier.

## 5 Experiments

The experiment section aims to answer the following questions: **(Q1)** Is Diff-PGD/Diff-rPGD effective to generate adv-samples with higher realism? **(Q2)** Can Diff-PGD be easily applied to generate better style-customized adv-samples? **(Q3)** Can Diff-PGD be applied to physical world attacks? **(Q4)** Do adversarial samples generated by Diff-PGD show better properties like transferability and anti-purification ability?

**Datasets, Models, and Baselines.** We use the validation dataset of ImageNet [8] as our dataset to get some statistical results for global attacks and regional attacks. For the style-guided adv-sample generation, we use some cases from [12] but also collect more images and masks by ourselves. We use VGG-19 as the backbone network to compute the style distance. The main target classifier to be attacked across this paper is ResNet-50 [18], and we also use ResNet-101, ResNet18, and WideResNet(WRN) [51] for the transferability test. Besides convolutional-based networks, we also try our Diff-PGD on Vision Transformers like ViT-b [11] and BEiT-l [3], and the results are put in Appendix E. The diffusion model we adopt is the unconditional diffusion model pre-trained on

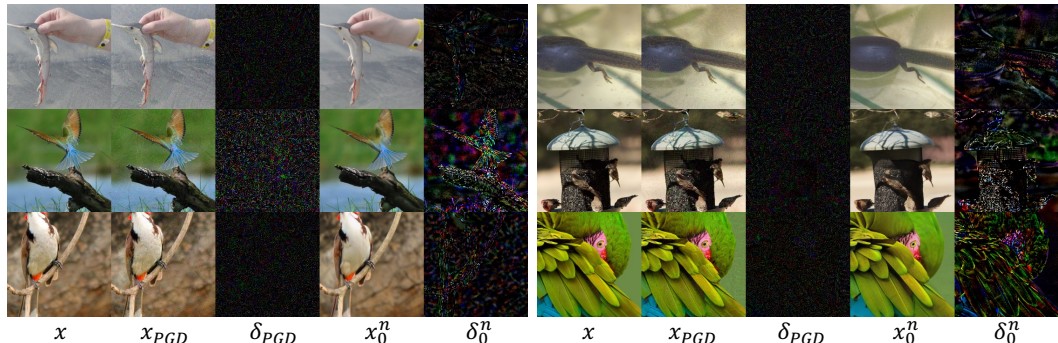

Figure 2: **Visualization of Adversarial Samples generated by Diff-PGD**: adv-samples generated using PGD ($x_{\text{PGD}}$) tend to be unnatural, while Diff-PGD ($x_0^n$) can preserve the authenticity of adv-samples. Here $x$ is the original image, $\delta_{\text{PGD}} = x - x_{\text{PGD}}$ and $\delta_0^n = x - x_0^n$, and we scale up the $\delta$ value by five times for better observation. Zoom in on a computer screen for better visualization.

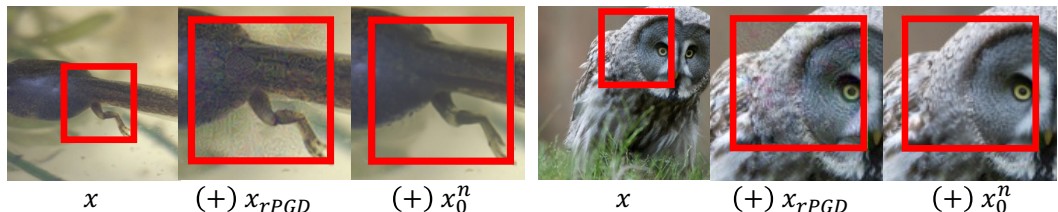

Figure 3: **Visualization of Adversarial Samples generated by Diff-rPGD**: Diff-rPGD can generate better regional attacks than PGD: the attacked region can better blend into the background. The attacked regions are defined using red bounding boxes, and $(+)$ means zoom-in.

ImageNet [9] though we use DDIM [40] to respace the original timesteps for faster inference. More details about the experimental settings are included in the appendix.

We compare Diff-PGD with threat models such as PGD [31] (for digital attacks), AdvPatch [4] (for physical-world attacks), and AdvCam [12] (for customized attacks and physical-world attacks). All threat models generate attacks within the context of image classification tasks. More details are included in the supplementary materials.

**Diff-PGD For Digital Attacks (Q1).** We begin with the basic global $\ell_\infty$ digital attacks, where we set $\ell_\infty = 16/255$ for PGD and Diff-PGD, and # of iterations $n = 10$ and step size $\eta = 2/255$. For Diff-PGD, we use DDIM with timestep $T_s = 50$ (noted as DDIM50 for simplicity), and $K = 3$ for the SDEdit module. Figure 2 shows that adv-samples generated by Diff-PGD ($x_0^n$) are more stealthy, while PGD-generated samples ($x_{PGD}$) have some patterns that can be easily detected by humans. This contrast is clearer on the perturbation map: $\delta_{PGD}$ contains more high-frequency noise that is weakly related to local patterns of the instance $x$, and $\delta_0^n$ is smoother and highly locally dependent. Similarly, for regional adversarial attacks, Diff-rPGD can generate adv-samples that can better blend into the unchanged region, as demonstrated in Figure 3, showing higher stealthiness than rPGD.

We also conduct experiments to show the effectiveness of Diff-PGD regarding the Success Attack Rate. We uniformly sampled 250 images from the ImageNet validation set. From Figure 6 (a), we can see that although the gradient is restricted to generate realistic perturbations, Diff-PGD can still reach a high success rate with more than 5 iterations. Detailed settings can be found in the supplementary materials.

We also conduct experiments on different norms. We show additional results of the performance of Diff-PGD on $\ell_2$-based attacks in Appendix F.

**Diff-PGD For Style-Customized Attacks (Q2).** For customized attacks, we focus on the task of generating adversarial samples using a mask $M$ and a style reference image $x_s$ provided by the user. We compare our approach, described in Section 4.3, with the AdvCam method. As previously mentioned, the main advantage of our pipeline is that we do not need to balance multiple losses, and can instead divide the task into two stages: generate a style-transferred image $\hat{x}_s$ without considering

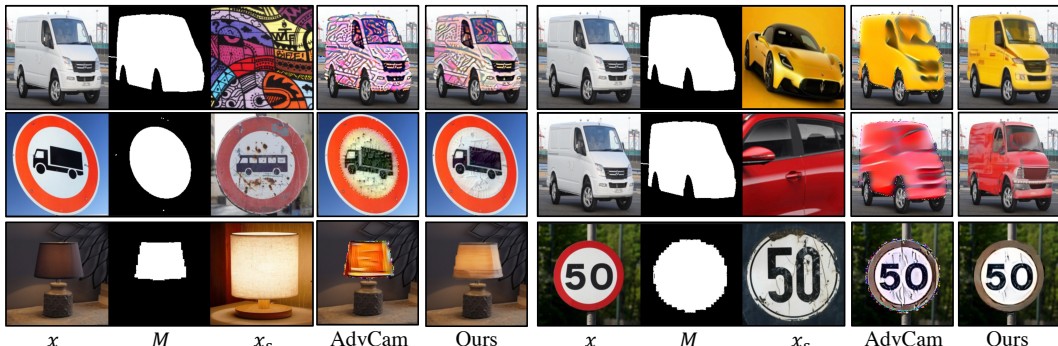

|   $x$   |   $M$   |   $x_s$   |   AdvCam   |   Ours   |   $x$   |   $M$   |   $x_s$   |   AdvCam   |   Ours   |

Figure 4: **Generating Adversarial Samples with Customized Style**: Given the original image $x$, a style mask $M$, and a style reference image $x_s$, Diff-PGD can generate more realistic samples, even in cases where only local styles are given (*e.g.* only the door of the red car is offered as a $x_s$).

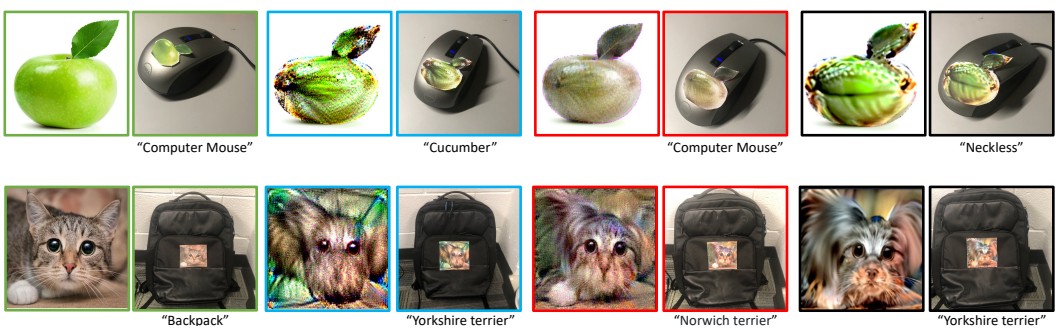

Figure 5: **Results of Physical-World Attacks**: We show two scenarios of physical world attacks: the first row includes untargeted attacks on a small object: computer mouse, and the second row includes targeted attacks on a larger object: backpack, where we set the target to be Yorkshire terrier. The sticks-photo pairs include clean patch (green box), AdvPatch(blue box), AdvCam generated patch (red box), and our Diff-Phys generated patch (black box).

adversarial properties, and then generate a realistic perturbation using Diff-PGD to make the image adversarial to the classifier.

For Diff-PGD, we use DDIM10 with $K = 2$. Figure 4 demonstrates that our method consistently generates customized adversarial samples with higher stealthiness, while the adversarial samples generated by AdvCam exhibit digital artifacts that make them appear unnatural. Also, in some cases, the style reference cannot be easily transferred (*e.g.* only local styles are given in $x_s$, or $x$ and $x_s$ are collected with different resolution), resulting in an unsatisfactory $\hat{x}_s$, such as red/yellow car style in Figure 4. Diff-PGD can refine the samples with more details, thanks to the strong generation capacity of diffusion models.

**Diff-PGD For Physical-World Attacks (Q3).** In the physical-world attacks, we set the problem as: given an image patch (start point of optimization), and a target object (*e.g.* attack a backpack), we need to optimize the patch to fool the classifier when attached to the target object. We use an iPhone 8-Plus to take images from the real world and use an HP DeskJet-2752 for color printing. In AdvPatch, only adversarial loss is used to optimize the patch; in AdvCAM, content loss and smooth loss are optimized together with adversarial loss; for Diff-Phys, we only use purified adversarial loss defined in Equation (8), and the SDEdit is set to be DDIM10 with $K = 2$ for better purification.

The results are shown in Figure 5, for the untargeted attack, where we use a green apple as a patch to attack a computer mouse. For targeted attack, we use an image of a cat to attack a backpack, with the target-label as "Yorkshire terrier". We can see that both AdvPatch and AdvCam generate patches with large noise, while Diff-Phys can generate smoother and more realistic patches that can successfully attack the given objects.

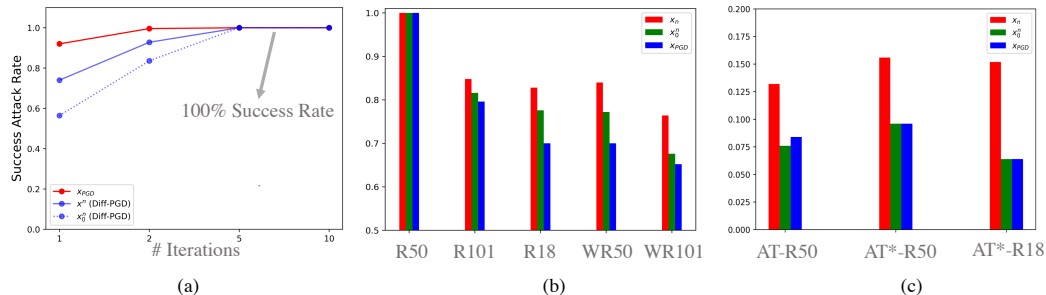

Figure 6: This figure presents quantitative results on our approach, with all y-axis representing the success rate of attacks: (a) Successful rate of Diff-PGD vs PGD; (b) Results of transferability of Diff-PGD vs PGD, where $\epsilon = 16/255$ and $\eta = 2/255$. The adv-samples are generated with ResNet-50 (R50) and tested on ResNet-18 (R18), ResNet-101 (R101), WRN-50 (WR50), and WRN-101 (WR101); (c) Success rate of Diff-PGD vs PGD on adversarially trained networks. AT uses adversarial training strategy in [13] and AT* uses AT strategy in [39] on the ImageNet dataset.

Table 2: **Anti-purification Results**: Adv-samples generated by PGD and Diff-PGD against adversarial purification: (+P) means the classifier is enhanced by an off-the-shelf adversarial purification module. Here we set $\epsilon = 16/255, n = 10$. The attacks are operated on ResNet-50.

| Sample | (+P)ResNet50 | (+P)ResNet101 | (+P)ResNet18 | (+P)WRN50 | (+P)WRN101 |
|---|---|---|---|---|---|
| $x_{\text{PGD}}$ | 0.35 | 0.18 | 0.26 | 0.20 | 0.17 |
| $x_n$ (Ours) | **0.88** | **0.38** | 0.36 | 0.32 | 0.28 |
| $x_n^0$ (Ours) | 0.72 | 0.36 | **0.37** | **0.36** | **0.36** |

**Exploring Other Properties of $x_0^n$ and $x^n$ (Q4).** Finally, we investigate additional properties of adversarial samples generated using Diff-PGD. Among the two samples $x_0^n$ and $x^n$, the former exhibits both adversarial characteristics and a sense of realism, while the latter is noisier but contains adversarial patterns that are more difficult for SDEdit to eliminate. Although our approach does not specifically target enhancing Transferability and Purification power, we demonstrate that Diff-PGD surpasses the original PGD in these two aspects.

**Transferability.** We test the adv-samples targeting ResNet-50 on four other classifiers: ResNet-101, ResNet-18, WRN-50 and WRN-101. From Figure 6(b) we can see that both $x^n$ and $x_0^n$ generated by Diff-PGD can be better transferred to attack other classifiers than PGD. This can be explained by the intuition that adv-attacks in semantic space can be better transferred. We also test the success rate attacking adversarially trained ResNet-50 in Figure 6(c) and we can see $x^n$ is much better than other adversarial samples.

**Anti-Purification.** Following the purification pipeline in [34], we attach an off-the-shelf sample purifier to the original ResNet-50 and test the success rate of different adv-samples. In Table 2 we can see that the adv-samples $x_{PGD}$ generated by PGD are easily purified using the enhanced ResNet-50. In contrast, our adv-samples, $x^n$ and $x_0^n$, show better results than $x_{PGD}$ by a large margin. It can be explained by the type of perturbations: out-of-distribution perturbations used in PGD can be removed using diffusion-based purification, while in-distribution attacks in Diff-PGD are more robust to purification.

## 6 Accelerated Diff-PGD with Gradient Approximation

From Algorithm 1, we can see that we need to calculate the derivative of the model's output over the input, where the model is composed of chained U-Nets $R_\phi$ in the diffusion model followed by a target classifier $f_\theta$. Here we focus on the computational bottleneck, which is the derivative of $K$-step SDEdit outputs $x_0$ over SDEdit input $x$:

$$\frac{\partial x_0}{\partial x} = \frac{\partial x_K}{\partial x}\left(\frac{\partial x_{K-1}}{\partial x_K}\frac{\partial x_{K-2}}{\partial x_{K-1}}...\frac{\partial x_1}{\partial x_0}\right) \approx c \qquad (10)$$

We approximate the gradient with a constant $c$. We notice that the approximation of this Jacobian has been used in many recent works [36, 49, 7]. We also tried this strategy in Diff-PGD, and we got the accelerated version:

$$\frac{\partial L(f(x_0^i))}{\partial x} \approx c \frac{\partial L(f(x_0^i))}{\partial x_0^n}. \tag{11}$$

From this we can see that, we only need to calculate the gradient over $x_0^n$! We only run the inference of SDEdit to get $x_0^n$ without saving the gradient (lower GPU memory, faster), making it much cheaper. We found that, for the global attack tasks, the approximated gradient still shows a high success rate but saves $50\%$ of time and $75\%$ of VRAM. More detailed results about the visualization and sucess rate can be found in Appendix C.

## 7 Conclusions

In this paper, we propose a novel method to power the generation of adversarial samples with diffusion models. The proposed Diff-PGD method can improve the stealthiness of adv-samples. We further show that Diff-PGD can be easily plugged into global digital attacks, regional digital attacks, customized attacks, and physical-world attacks. We demonstrate through experiments that our methods outperformed the baselines and are effective and stable. The major limitation of our method is that the back-propagation is more expensive than traditional gradient-based methods; however, we believe that the strong prior knowledge of the diffusion model can push forward adversarial attacks & defenses. Finally, while the proposed adversarial attack method could be potentially used by malicious users, it can also enhance future efforts to develop robust defense mechanisms, thereby safeguarding the security of AI systems.

## Acknowledgment

The authors would like to thank the anonymous reviewers for useful comments. HX and YC are supported by grants NSF 2008513 and NSF 2206576. AA is supported in part by the Army Research Office under grant number W911NF-21-1-0155 and by the New York University Abu Dhabi (NYUAD) Center for Artificial Intelligence and Robotics, funded by Tamkeen under the NYUAD Research Institute Award CG010. BH is supported by the NSF award CAREER-2048168 and the AFOSR award FA9550-23-1-0732.

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

# Contents

# Appendix

The appendix is organized as follows: in Section A, we provide a summary table of notations to assist readers in better understanding the content. Then we delve into the details of our algorithms in Section B, including how to apply Diff-PGD to style-based attacks and physical world attacks step by step. Then, we show the experimental settings in Section D, including our experimental environments, training settings, and inference speed. Then, we show more complementary results of our experiments in Section G, where we also present the results of a human-evaluation survey. Finally, we discuss the potential social impacts of our work in Section I.

## A  Notations

| Notation | Description |
|---|---|
| $p(x_0)$ | real data distribution |
| $q(x_t|x_0)$ | forward diffusion in diffusion model |
| $p_\phi(x_{t-1}|x_t)$ | backward diffusion parameterized by $\phi$ |
| $\alpha, \beta$ | diffusion coefficients in a diffusion model |
| $R_\phi$ | backward diffusion sampling |
| $T$ | original diffusion time step |
| $K$ | SDEdit reverse timesteps |
| $n$ | iterations of PGD attack |
| $\epsilon$ | maximum $\ell_\infty$ norm of adversarial attacks |
| $\eta$ | steps of adversarial attacks |
| $\mathcal{B}_{\infty,\epsilon}$ | $\ell_\infty$ ball sized $\epsilon$ |
| $x^n$ | Diff-PGD generated adv-samples (before purification) |
| $x_0^n$ | Diff-PGD generated adv-samples (after purification) |
| $x_{\text{PGD}}$ | PGD-generated adv-samples |
| $x_{\text{rPGD}}$ | regional PGD-generated adv-samples |
| $x_s$ | style reference image in style-based attacks |
| $\hat{x}_s$ | style-transferred image |
| $x_c$ | copy of the original image, no gradient |
| $H_s$ | style feature extractor |
| $H_c$ | content feature extractor |
| $M$ | mask to define the regional attacks |
| $T_s$ | DDIM timestep |
| $K_s$ | SDEdit reverse timesteps with accelerator |
| $\circ$ | element-wise multiplication |
| $U$ | uniform distribution |

## B  Details about Methods

### B.1  Details about Style Diff-PGD

Departing from existing methods where adversarial loss and style loss are optimized together, Diff-PGD makes it possible to separate the optimization of these two losses, as Diff-PGD can guarantee the realism of output samples during the adversarial optimization stage. The detailed implementation is demonstrated in Algorithm 2, where we add a stage for style-based editing to Diff-PGD.

Even though here we only present one direction of customized editing (*e.g.* text-guided editing), this framework can deal with almost any customized adv-sample generation problem, by adopting any proper method for the first stage.

### B.2  Details about Physical World Diff-PGD

**Algorithm 2** Diff-PGD for Style-based Attacks
___
**Require:** Target classifier $f_\theta$, original image $x$, mask $M$, style reference $x_s$, style network $f_h$, style extract layers $H_s$, content extract layers $H_c$, denoised sampler $R_\phi$, style transfer learning rate $\eta_s$, # of reverse SDEdit steps $K$, iterations for PGD $n$, iteration for style transfer $n_s$, stepsize $\eta$, clip value $\epsilon$, style weight $\lambda_s$ and content weight $\lambda_c$
$x_s^0 = x$
**for** $t = 0, 1, 2, \ldots, n_s - 1$ **do**
  $L_s = \sum_{h \in H_s} \|G(f_h(x_s^t)) - G(f_h(x_s))\|_2^2$
  $L_c = \sum_{h \in H_c} \|f_h(x_s^t) - f_h(x_s)\|_2^2$
  $x_s^{t+1} = x_s^t - \eta_s \nabla_{x_s^t}(\lambda_s L_s + \lambda_c L_c)$                  ▷ Style transfer only, without adversarial loss
**end for**
$x^0 = x_0^0 = x_s^{n_s}$
$x_c = x_s^{n_s}$                                                           ▷ Used for unchanged part
**for** $t = 0, 1, 2, \ldots, n - 1$ **do**
  $x_K^t \sim q(x_K^t | x^t)$                        ▷ Sample $x_K^t$ from $q(x_K^t | x^t)$ in each PGD iteration
  **for** $i = K - 1, \ldots, 0$ **do**
    $x_i^t = R_\phi(x_{i+1}^t)$                ▷ Apply denoiser $R_\phi$ to $x_{i+1}^t$ in each SDEdit iteration
    $x_i^t \sim M \circ x_i^t + (1 - M) \circ q(x_i^t | x^t)$ ▷ Sample from masked combination of $x_i^t$ and $q(x_i^t | x^t)$
  **end for**
  $g = \nabla_{x^t} l(f_\theta(M \circ x_0^t + (1 - M) \circ x_c)]$              ▷ Compute the gradient
  $x^{t+1} = \Pi_{x,\epsilon}[x^t + \eta M \circ \text{sign}(g)]$                     ▷ PGD update
**end for**
$x_0^n = \text{rSDEdit}(x^n)$                             ▷ Apply reverse SDEdit to the final $x^n$
**return** $x_0^n$                                   ▷ Return style-based adversarial sample
___

**Algorithm 3** Diff-PGD for Physical World Attacks
___
**Require:** Target classifier $f_\theta$, original patch image $x$, mask $M$, denoised sampler $R_\phi$, # of reverse SDEdit steps $K$, iterations $n$, stepsize $\eta$, physics transformation set $\mathcal{T}$, learning rate $\eta$
$x^0 = x_0^0 = x$, $x_c = x$
**for** $t = 0, 1, 2, \ldots, n - 1$ **do**
  $x_K^t \sim q(x_K^t | x^t)$                        ▷ Sample $x_K^t$ from $q(x_K^t | x^t)$ in each PGD iteration
  **for** $i = K - 1, \ldots, 0$ **do**
    $x_i^t = R_\phi(x_{i+1}^t)$                ▷ Apply denoiser $R_\phi$ to $x_{i+1}^t$ in each SDEdit iteration
    $x_i^t \sim M \circ x_i^t + (1 - M) \circ q(x_i^t | x^t)$ ▷ Sample from masked combination of $x_i^t$ and $q(x_i^t | x^t)$
  **end for**
  $\tau \sim U(\mathcal{T})$                                       ▷ Sample physics transformation
  $g = \nabla_{x^t} l(f_\theta(\tau(M \circ x_0^t + (1 - M) \circ x_c))]$         ▷ Compute the gradient
  $x^{t+1} = x^t + \eta M \circ \text{sign}(g)$                             ▷ update
**end for**
$x_0^n = \text{rSDEdit}(x^n)$                             ▷ Apply reverse SDEdit to the final $x^n$
**return** $x_0^n$                                     ▷ Return Diff-Phys patch
___

To adapt Diff-PGD to the physical world settings, larger modifications are needed. In Algorithm 3, we present Diff-PGD in physical world environments. The biggest difference is that we no longer restrict the $\ell_\infty$ norm of perturbations. Also, we use gradient descent instead of projected gradient descent. For the generation of the physical adaptor $\mathcal{T}$, we consider the scale of the patch image $x$, the position of the patch image, the background images, and the brightness of the environment.

## C   Accelerated Diff-rPGD

Changing only one line can help us get the accelerated version of generalized Diff-PGD with an approximated gradient (Algorithm 4).

**Algorithm 4** Accelerated Diff-rPGD

---

**Require:** Target classifier $f_\theta$, original image $x$, mask $M$, denoiser $D_\phi$, # of reverse SDEdit steps $K$, iterations $n$, stepsize $\eta$, clip value $\epsilon$ (when $M = 1$ it reduces to Diff-PGD)

$x^0 = x_0^0 = x$, $x_c = x$

**for** $t = 0, 1, 2, \ldots, n - 1$ **do**

  $x_K^t \sim q(x_K^t | x^t)$                    ▷ Sample $x_K^t$ from $q(x_K^t | x^t)$ in each PGD iteration

  **for** $i = K - 1, \ldots, 0$ **do**

    $x_i^t = R_\phi(x_{i+1}^t)$                ▷ Apply denoiser $R_\phi$ to $x_{i+1}^t$ in each SDEdit iteration

    $x_i^t \sim M \circ x_i^t + (1 - M) \circ q(x_i^t | x^t)$ ▷ Sample from masked combination of $x_i^t$ and $q(x_i^t | x^t)$

  **end for**

  $g = \nabla_{x_0^t} l[f_\theta(M \circ x_0^t + (1 - M) \circ x_c)]$           ▷ Compute the approximated gradient

  $x^{t+1} = \Pi_{x, \epsilon}[x^t + \eta \, M \circ \text{sign}(g)]$                     ▷ PGD update

**end for**

$x_0^n = \text{rSDEdit}(x^n)$                     ▷ Apply reverse SDEdit to the final $x^n$

**return** $x^n, x_0^n$          ▷ Return the final adversarial example and the denoised version

| Method | $K$ | $n$ | VRAM(G) | Speed(sec/sample) |
|---|---|---|---|---|
| Diff-PGD | 2 | 10 | ∼18 | 8 |
| v2 | 2 | 10 | ∼4 | 4 |
| Diff-PGD | 3 | 10 | ∼20 | 10 |
| v2 | 3 | 10 | ∼4 | 5 |

Table 3: The Speed and VRAM of Diff-PGD and accelerated Diff-PGD with approximal gradient (v2). From this we can see that the v2 is much cheaper to run than the original Diff-PGD. $K$ and $n$ follows the definition in Algorithm 1.

Here, we compared the speed and success rate of accelerated Diff-PGD with the original version; we refer to the accelerated version as v2. The qualitative results of v2 are put in Figure 10, and the quantitative results about speed, VRAM, and success rate are put in Table 3 and Table 4.

# D Implementation Details

**Experimental Environments and Inference Speed** The methods described in this study are implemented using the PyTorch framework. All the experiments conducted in this research were carried out on a single RTX-A6000 GPU, housed within a Ubuntu 20.04 server.

Since we need to do back-propagation on the U-Net used in the diffusion model, it will cost more GPU memory and GPU time. Running on one single GPU, it takes $\sim 7$ seconds to run Diff-PGD with $n = 10$, $K_s = 2$ for one sample on one A6000 GPU. Parallel computing with more GPUs can be used to accelerate the speed.

| Method | Model | $n$ | $\epsilon$ | SR |
|---|---|---|---|---|
| v2 | ResNet-50 | 10 | 16/255 | 99.6 |
| v2 | ResNet-50 | 10 | 8/255 | 98.8 |
| v2 | ResNet-50 | 15 | 16/255 | 100.0 |
| v2 | ResNet-50 | 15 | 8/255 | 100.0 |

Table 4: The success rate of accelerated Diff-PGD, from which we can see the success rate (SR) is still really nearly $100\%$, showing that v2 is a cheaper but still effective attack. $\epsilon$ and $n$ follows the definition in Algorithm 1.

**Global Attacks** Here we use $\epsilon = 16/255, \eta = 2/255, n = 10$ as our major settings (except for the ablation study settings) for both PGD and Diff-PGD. For Diff-PGD, we use DDIM50 with $K_s = 3$ as our SDEdit setting; we will also show more results with different DDIM time-steps and different $K_s$.

**Regional Attacks** For the experiments of the regional attacks, we randomly select a squared mask sized $0.4H \times 0.4W$, where the size of the original image is $H \times W$. For the same image in the ImageNet dataset, we use the same mask for both rPGD and Diff-rPGD. Also, we use $\epsilon = 16/255, \eta = 2/255, n = 10$ for bounded attack with projected gradient descent. And for Diff-rPGD, we use DDIM50 with $K_s = 2$; similarly, we try more settings in the ablation study section. The repainting strategy of our Diff-rPGD is based on replacement, but more effective strategies are encouraged to be used here.

**Style-based Attacks** As mentioned in Section B, the implementation of Diff-PGD for style-based attacks can be divided into two parts. For the first stage, we use style weight $\lambda_s = 4000$ and adopt Adam optimizer with learning rate $\eta = 0.01$ for style transfer. For the feature extractor, we adopt VGG-19 as our backbone with the first five convolutional layers as the style extractor $H_s$ and the fourth convolutional layer as the content extractor $H_c$. For the target image $x$, we use SAM to collect the segmentation mask to define the attack region.

In the second stage, we utilize DDIM10 with a parameter setting of $K_s = 2$ to achieve better purification. It is beneficial to use a smaller $K_s$, as the style-transferred images often contain unwanted noise and art-styled patterns that can be mitigated through this approach.

**Phyiscal World Attacks** The background images in the physical world attacks are collected by taking photos of the target object with different camera poses. For the physical adaptor, we set the scale factor to be around $[0.8, 1.2]$ of the original patch size; then we set the random brightness to be $[0.5, 1.5]$ of the original brightness. We also set a margin for the random positions of the image patch so that the patch will not be too far away from the target object. We use an iPhone 8-Plus to take images from the real world and use an HP DeskJet-2752 for color printing. For Diff-PGD we use DDIM10 and $K_s = 2$ to allow larger purification.

**Transferability and Anti-Purification Power** We show results of two additional good properties of $x^n$ and $x_0^n$ generated by Diff-PGD: transferability and anti-purification power. For transferability, we use $\epsilon = 16/255, \eta = 2/255, n = 10$ in the main paper, with DDIM50 with $K_s = 2$ for Diff-PGD. Targeting our network on ResNet-50, we test the adv-samples among ResNet-101, ResNet18, and WRN-50 and WRN-101.

For the test of anti-purification power, we use an SDEdit with DDIM50 with a larger $K_s = 5$ and test for both PGD and Diff-PGD with $\epsilon = 16/255, \eta = 2/255, n = 10$ and DDIM50 with $K_s = 2$ for Diff-PGD so that our method is not biased.

## E    Results on Vision Transformers

Compared with the Convolutional-based network such as ResNet we used before, Vision Transformers are another branch of strong classifiers that are worth taking into consideration. Here we try a global attack on two famous models on ImageNet, ViT-b (https://huggingface.co/google/vit-base-patch16-224) and BEiT (https://huggingface.co/microsoft/beit-large-patch16-224-pt22k-ft22k).

Figure 9 demonstrates that Diff-PGD can still work well to generate adversarial samples with higher stealthiness. Table 5 shows that Diff-PGD is still effective for vision transformers.

## F    Results on $\ell_2$-based Attacks

We also conduct experiments for $\ell_2$-based attacks, we set $n = 10$ and $\epsilon(\ell_2) = 16/255$, and we can still get a high success rate as $100\%$ and the generated adversarial samples are shown in Figure 8, which turns out to be also realistic.

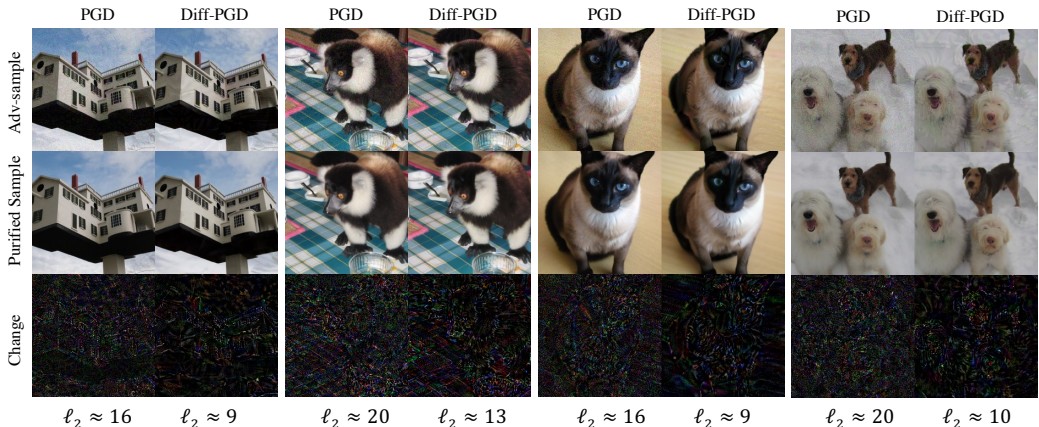

Figure 7: **Purification on Adv-samples Generated by PGD vs Diff-PGD**: PGD attack can be easily purified using recently proposed purification strategies using diffusion model, while Diff-PGD can generate adv-samples in the realistic domain, making it harder to be purified. The changes in PGD contain a lot of noise while the changes of Diff-PGD contain more meaningful features, which means that the adversarial noise can be easily removed for traditional PGD. Also, we show the $\ell_2$ norm of the changes, from which we can see that changes of Diff-PGD are always smaller.

| Model | $n$ | $\epsilon$ | SR |
|---|---|---|---|
| ViT-base | 10 | 8/255 | 100 |
| ViT-base | 10 | 16/255 | 100 |
| BEiT-large | 10 | 8/255 | 99.6 |
| BEiT-large | 10 | 16/255 | 99.6 |

Table 5: Success rate of Diff-PGD on vision transformers.

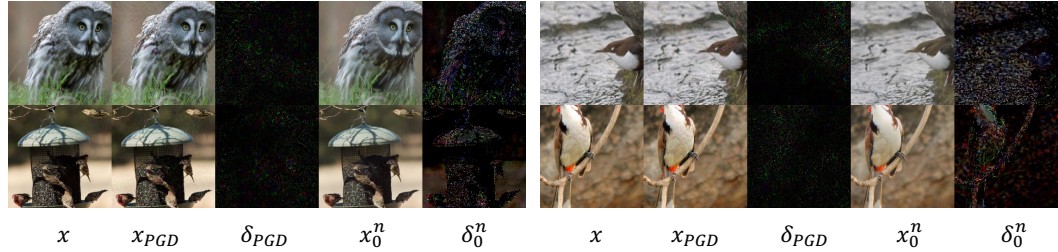

Figure 8: **Performance of Diff-PGD on $\ell_2$ perturbations** : all the figures attack ResNet-50 with $\epsilon = 16/255$ under $l_2$ norm. (Zoom-in for better observation)

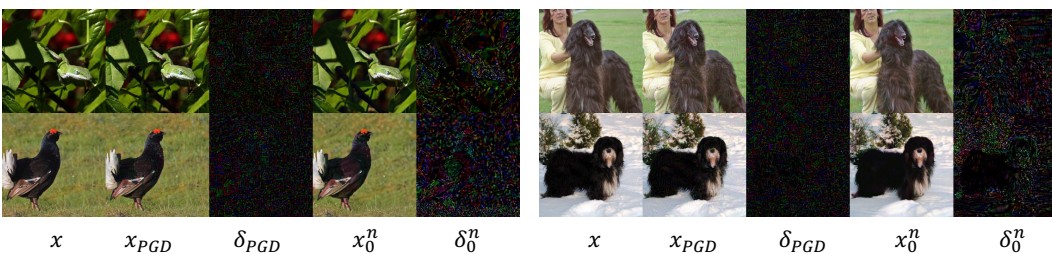

Figure 9: **Qualitative Results of Diff-PGD on Vision Transformers**: the left-half attack ViT-b and the right-half attack BEiT-l. (Zoom-in for better observation)

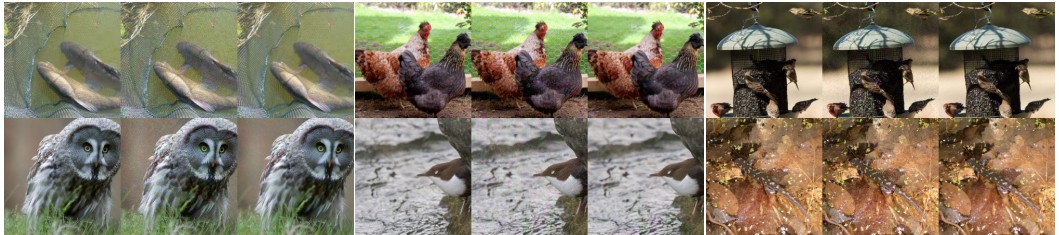

Figure 10: **Qualitative Results of Diff-PGD with Gradient Approximation** : here we show six samples, each contains $x, x^n, x_0^n$ from left to right. (Zoom-in for better observation)

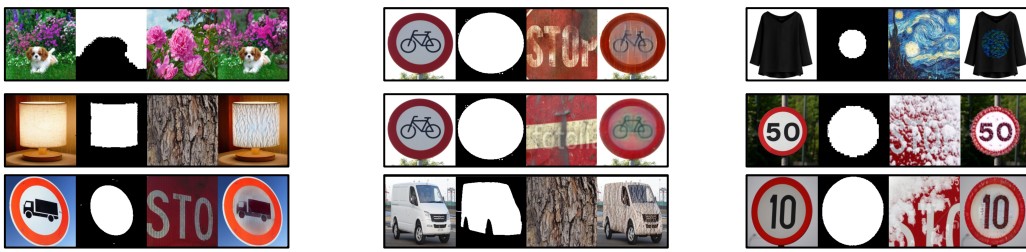

Figure 11: **More Results of Style-based Attacks:** we show nine samples from the constructed dataset, each sample contains $x, M, x_s, x_0^n$ from left to right.

## G    More Complementary Results

**Global Attacks**    Here we present more results randomly sampled from the ImageNet validation set to show that Diff-PGD can steadily generate adv-samples with high stealthiness. In Figure 12, we show some complementary results of $\epsilon = 16/255, \eta = 2/255, n = 10$ in main part of the paper. Also in Figure 13, we show results when the perturbation bound is larger: $\epsilon = 32/255, \eta = 4/255, n = 10$, where we use DDIM10 with $K_s = 2$; from the figure we can see that Diff-PGD can still generate realistic adv-samples.

Also, we demonstrate that Diff-PGD remains effective in generating adversarial samples with a high success rate even when the attack bound is smaller. Figure 14 illustrates this point, where we set $\epsilon = 8/255$ and evaluate two Diff-PGD strategies: DDIM50 and DDIM100, all with $T_s = 2$. Both strategies exhibit a notable success rate and can produce realistic adversarial samples.

**Regional Attacks**    Diff-rPGD can be adopted in customized settings where only masked regions can be attacked. Here, we show more results of regional attacks in Figure 20, where $\epsilon = 16/255, \eta = 2/255, n = 10$. We compare Diff-rPGD with rPGD, and the results show that our Diff-rPGD can deal with the regional adversarial attack better: the masked region can be better merged into the background.

**Style-based Attacks**    In the main paper, we show that Diff-PGD can help generate style-based adv-samples effectively. In Figure 15, we further present that the second stage in our two-stage strategy is necessary and can highly preserve the realism of the output images.

**Phyiscal World Attacks**    We present the result of physical world attacks using adversarial patches generated using Diff-PGD in the main paper. In order to show that the adversarial patches are robust to camera views, we show more images taken from different camera views in Figure 16. For the two cases: computer mouse (untargeted) and back bag (targeted to Yorkshire Terrier), we randomly sample ten other camera views. The results show that the adversarial patches are robust both for targeted settings and untargeted settings. For targteted settings, our adversarial patch can fool the network to predict back bag as terriers, and for untargeted settings, the adversarial patch misleads the network to predict computer mouse as artichoke.

**Transferability and Anti-Purification**    We presented the results when $\epsilon = 16/255$ and found that: though designed for improved stealthiness and controllability, it is surprising that Diff-PGD transfers better than the original PGD to five different networks, here we show more results on different $\ell_\infty$

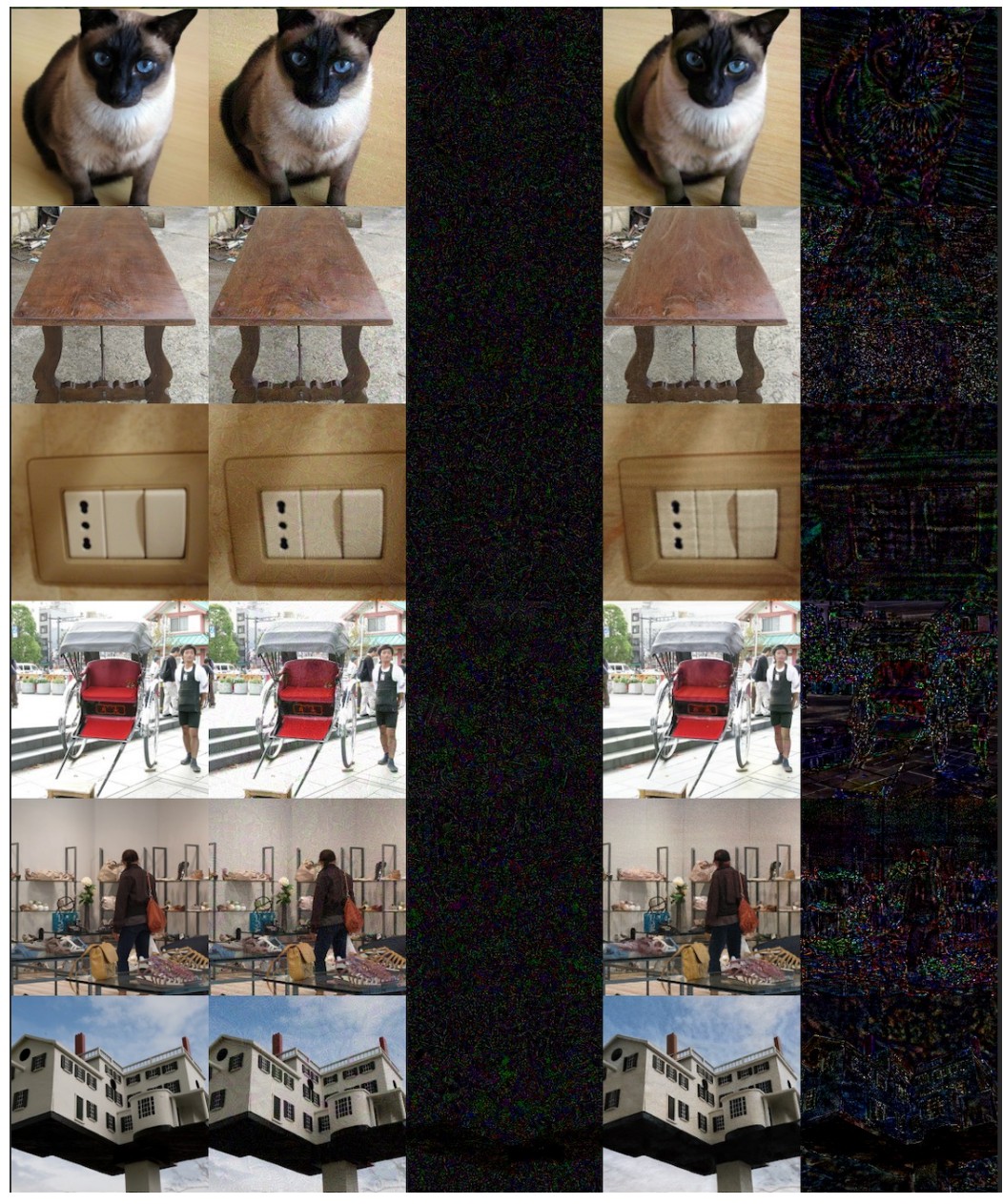

Figure 12: **More Results of Diff-PGD with** $\epsilon = 16/255$: five columns of each image block follows $x$, $x_{PGD}$, $\delta_{PGD}$, $x_0^n$ and $\delta_0^n$ as is defined in the main paper. We can see that Diff-PGD can steadily generate adv-samples with higher stealthiness. Zoom in on a computer screen for better visualization.

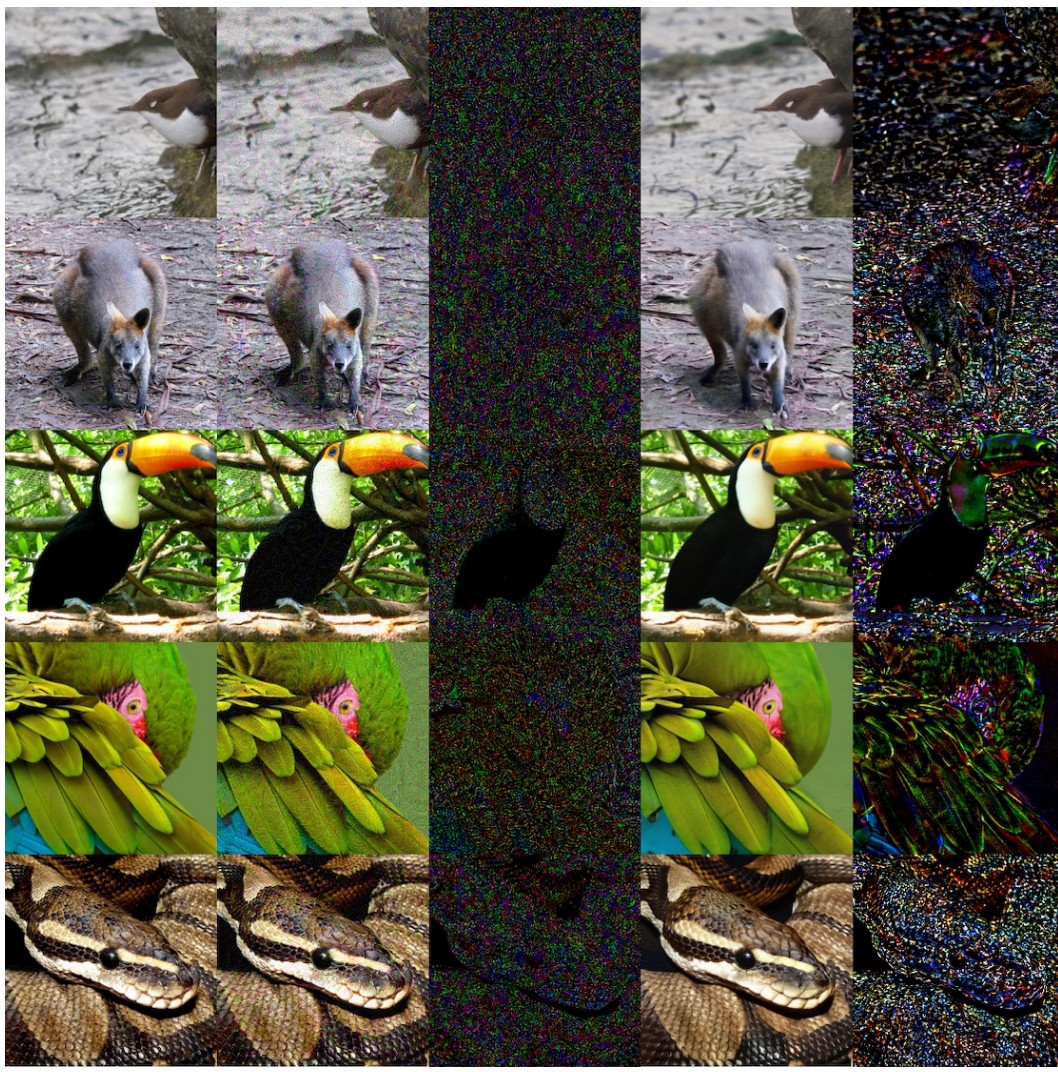

Figure 13: **More Results of Diff-PGD with $\epsilon = 32/255$**: five columns of each image block represent $x$, $x_{PGD}$, $\delta_{PGD}$, $x_0^n$ and $\delta_0^n$ respectively. We can see that Diff-PGD can steadily generate adv-samples with higher stealthiness. Zoom in on a computer screen for better visualization.

bound. We take the other two settings: $\epsilon = 8/255$ and $\epsilon = 32/255$ to better investigate our methods, for $\epsilon = 8/255$ we still use $\eta = 2/255$ while for $\epsilon = 32/255$, we use $\eta = 4/255$ for better attacks, both of them use $n = 10$. For Diff-PGD, we use DDIM50 with $K_s = 2$ for $\epsilon = 8/255$ case and DDIM30 with $K_s = 2$ for $\epsilon = 32/255$ case.

Results shown in Figure 17 reveal that both $x_0^n$ and $x^n$ exhibit superior transferability compared to the original PGD attack. This observation aligns with our intuition that attacks operating within the realistic domain are more likely to successfully transfer across different networks.

For anti-purification, we show some qualitative results that how the noises can be removed by running SDEdit on PGD and Diff-PGD. In Figure 17, we show visualizations of the adv-samples, purified adv-samples, and the changes during the purification. From the figure, we can find that the adversarial patterns generated by PGD can be easily removed by diffusion-based purification and Diff-PGD.

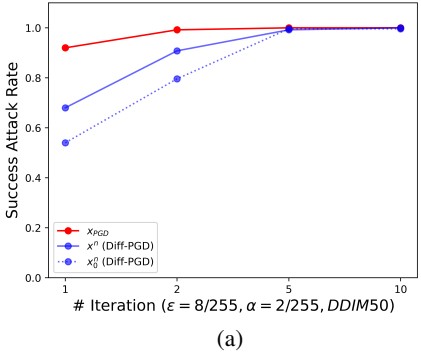
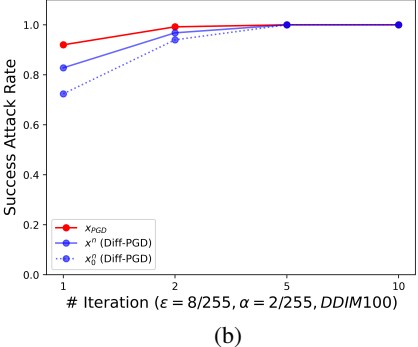

(a)                                              (b)

Figure 14: **Effective of Diff-PGD with Smaller Bound**: we show that when the attack $\ell_\infty$ bound is smaller: $\epsilon = 8/255$, Diff-PGD is still effective to generate adv-samples with high success rate. (a) we use DDIM100 and $K_s = 2$, (b) we use we use DDIM50 and $K_s = 2$.

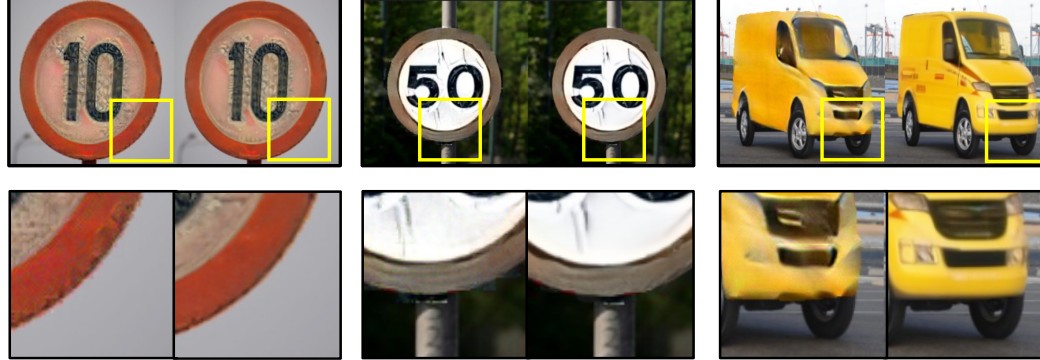

Figure 15: **Results of Style-based Adv-sample Generation in Two Stages:** here we should three cases in style-based adversarial attacks, for each case, we should the difference between $\hat{x}_s$(the left image in each case) and $x_0^n$(the right image in each case), which demonstrate that by employing Diff-PGD, we are able to generate adversarial samples while simultaneously mitigating the artifacts that arise during the style transfer stage.

## H   Human Evaluation

To better evaluate the stealthiness of Diff-PGD compared with the original PGD attack, we conduct a survey among humans with the assistance of Google Form. The user interface of the survey is shown in Figure 21, where the participants are allowed to choose multiple candidate images that they think are realistic. We randomly choose ten images from the ImageNet validation set and the choices for each question include one clean image, one adv-sample generated by PGD, and one adv-sample generated by Diff-PGD. For Diff-PGD we use DDIM50 with $K_s = 3$, and for both Diff-PGD and PGD we use $\epsilon = 16/255, \eta = 2/255, n = 10$.

We collect surveys from 87 participants up to the completion of the writing and most of them are not familiar with adversarial attacks. Figure 18 illustrates the results, indicating that Diff-PGD approach achieves higher stealthiness among human participants compared with PGD.

## I   Potential Social Impacts

Adversarial attacks pose a serious threat to the security of AI systems. Diff-PGD introduces a novel attack technique that effectively preserves the realism of adversarial samples through the utilization of the diffusion model. This holds true for both digital and real-world settings. The emergence of

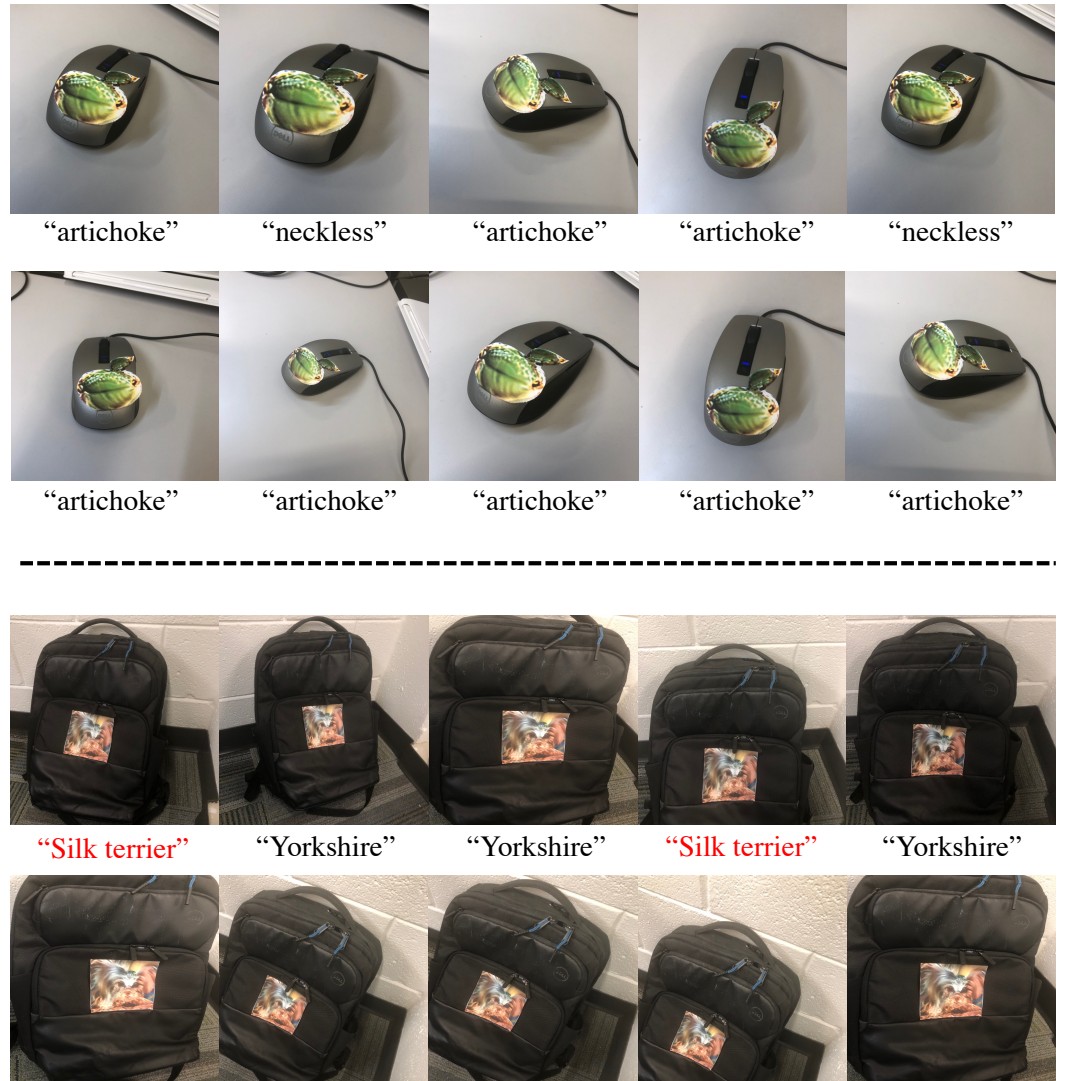

Figure 16: **Physical World Attack with Diff-PGD:** to show the effectiveness of adversarial patches generated by Diff-PGD, we show the images captured from different camera viewpoints for the two cases: computer mouse (untargeted) and back bag (targeted). For each case, we randomly sample other ten poses. We present the prediction of the target network under each image, where Yorkshire is short for Yorkshire terrier.

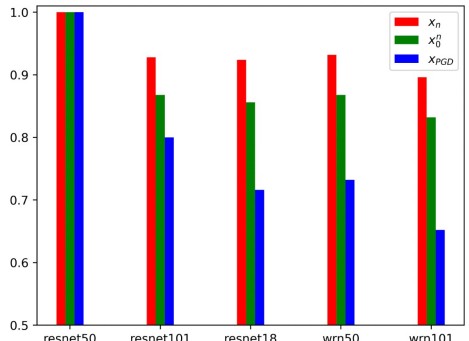 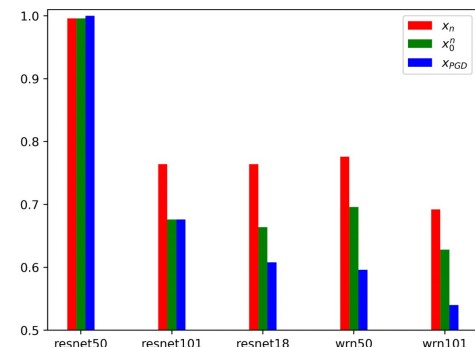

Figure 17: **More Results on Transferability Test**: in the main paper we show results of transferability of $\epsilon = 16/255$ settings; here we test transferability of Diff-PGD under different $\ell_\infty$ bounds, (Left): $\epsilon = 32/255, \eta = 4/255, n = 10$, for Diff-PGD we use DDIM50 with $K_s = 2$, (Right): $\epsilon = 8/255, \eta = 2/255, n = 10$ and for Diff-PGD we use DDIM30 with $K_s = 2$. From the plots we can see that Diff-PGD has better transferability power than PGD.

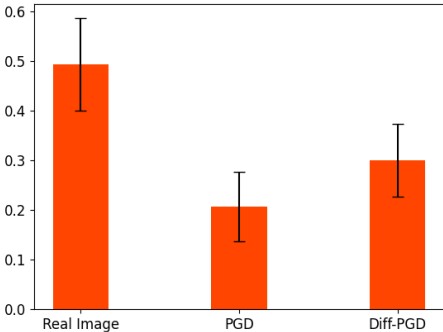

Figure 18: **Human Evaluation**: Collecting 87 samples using Google Form, we calculate the average choose rate of each kind of sample (real images, PGD-attacked images, and Diff-PGD-attacked images). The results show that Diff-PGD method exhibits greater stealthiness than PGD when evaluated by human participants.

such realistic domain attacks demands our attention and necessitates the development of new defense methods.

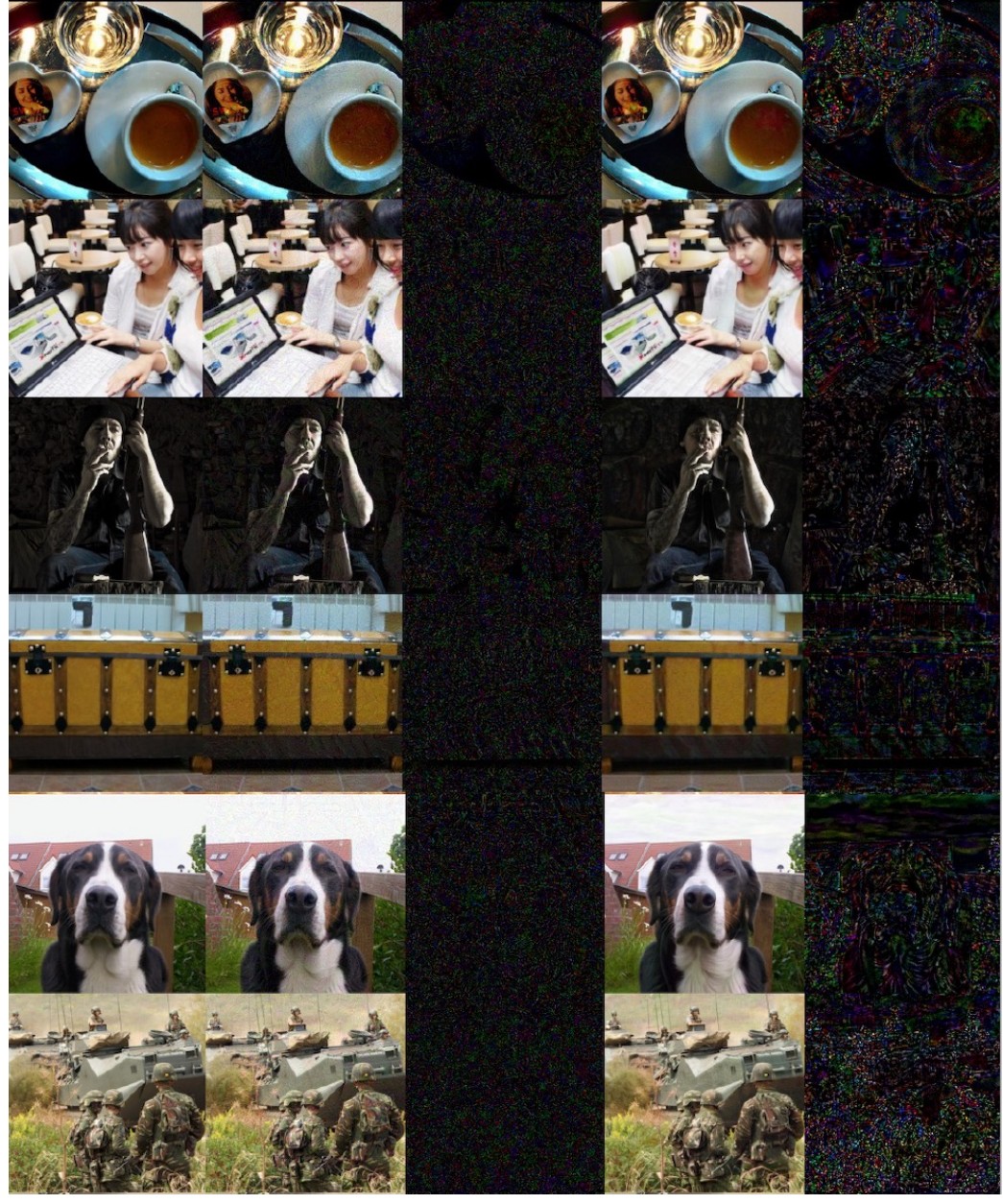

Figure 19: **More Results of Diff-PGD with** $\epsilon = \mathbf{16/255}$: five columns of each image block follows $x$, $x_{PGD}$, $\delta_{PGD}$, $x_0^n$ and $\delta_0^n$ as is defined in the main paper.

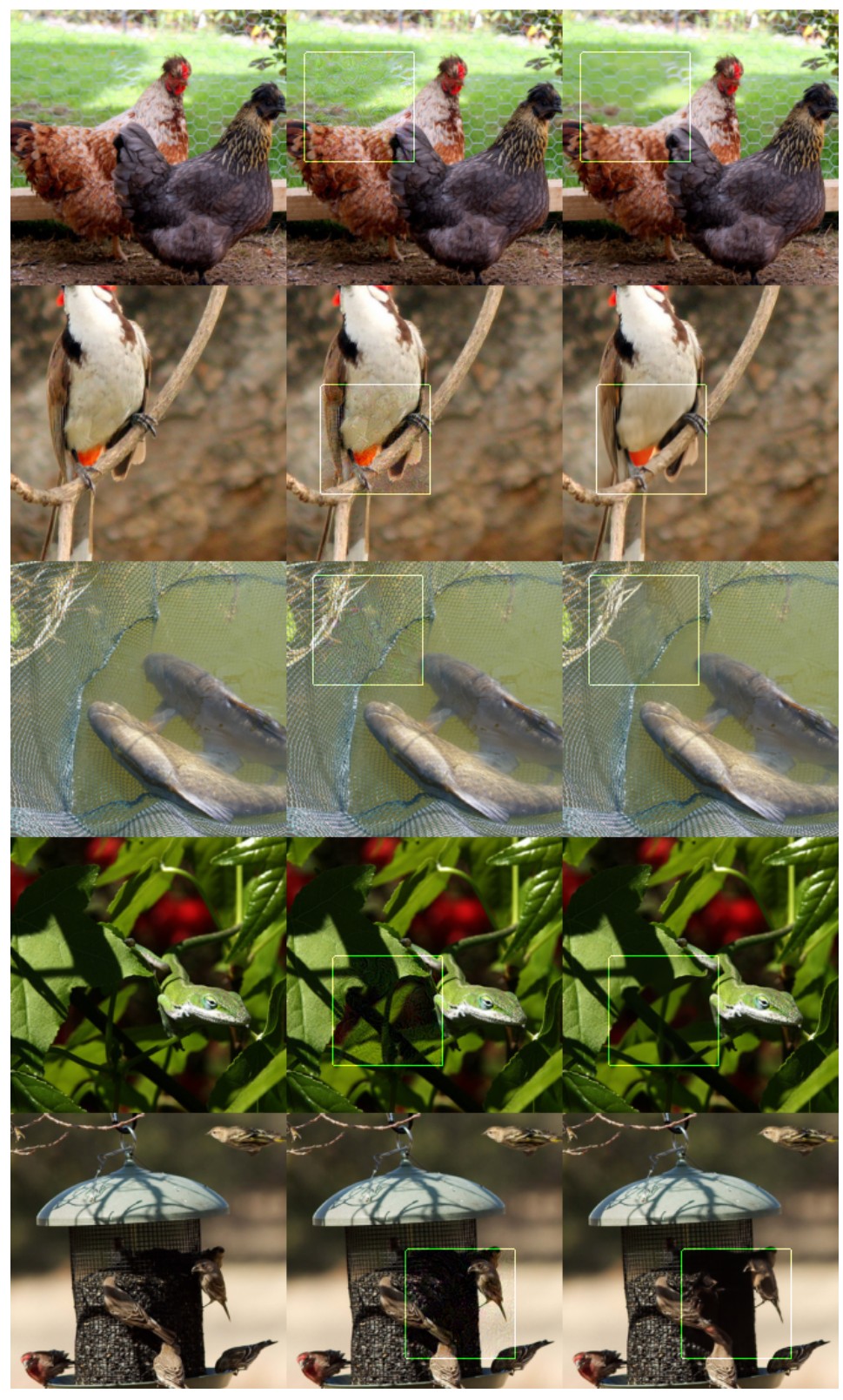

Figure 20: **More Regional Attack Results**: we present more results for regional attacks, the three columns are clean image, $x_{\text{rPGD}}$ and $x_0^n$ respectively.

Figure 21: **The Interface of our Survey for Human-Evaluations**: one sample of our questions is presented, and we randomized the order of the three images for all questions. The options consisted of a clean image, a PGD-attacked image, and a Diff-PGD-attacked image, with the participants unaware of the composition of the choices.

