# OpenReview forum: "Diffusion-Based Adversarial Sample Generation for Improved Stealthiness and Controllability"
_NeurIPS.cc/2023/Conference — NeurIPS 2023 poster_

### Official Review · Reviewer_PvL7 · 2023-06-29

**Soundness:** 3 good
**Presentation:** 3 good
**Contribution:** 3 good
**Rating:** 7
**Confidence:** 4

**Summary:**

The paper proposes a framework called Diffusion-Based Projected Gradient Descent (Diff-PGD) for generating realistic and stealthy adversarial samples to deceive neural network models. Diff-PGD utilizes a diffusion model to guide the optimization process, ensuring that adversarial samples remain close to the original data distribution while maintaining their effectiveness. The framework offers customization for specific attack scenarios, such as digital attacks, physical-world attacks, and style-based attacks. It separates the optimization of the adversarial loss from other surrogate losses, leading to improved stability and controllability. The authors demonstrate that adversarial samples generated using Diff-PGD exhibit better transferability and anti-purification power compared to traditional gradient-based methods.

**Strengths:**

## Originality
(+) The paper introduces a novel framework, Diff-PGD, which combines the use of diffusion models with adversarial sample generation. This integration of diffusion models into the generation of adversarial samples is a unique approach that has not been explored extensively in prior research.

## Quality
(+) The paper provides a thorough description of the proposed framework, Diff-PGD, and its underlying principles.

## Clarity
(+) The paper is well-written and effectively communicates the key concepts and ideas.

## Significance
(+) By introducing the Diff-PGD framework, the authors offer a solution that improves the stealthiness of adversarial samples while maintaining their effectiveness.

(+) The provided experimental evaluations (quantitative and qualitative) are convincing.


**Weaknesses:**

(-) The phenomenon of the vulnerability of DNNs/ViTs to AEs is well known. How does the proposed attack perform against hardened models, i.e. adversarially trained models?
(-) Did the authors experiment with transformer-based models, such as ViT? I would be interested in the attack performance of these models as well.


**Questions:**

Please address my questions in the weakness section.


**Limitations:**

The authors elaborated on the limitations of the approach.

---

> ### Author Rebuttal · Authors · 2023-08-10
>
> We express our gratitude to the reviewer for conducting a careful review and providing valuable feedback. We highly appreciate the reviewer's recognition of the effectiveness of our method. In this response, we address the questions raised by the reviewer:
>
>
>
> > The phenomenon of the vulnerability of DNNs/ViTs to AEs is well known. How does the proposed attack perform against hardened models, i.e. adversarially trained models?
>
>
>
> In Figure 6(c), we compare the success rate of Diff-PGD vs PGD on adversarially trained networks, where the samples are generated on the **naturally-trained ResNet-50**.
>
>
>
> We conduct further experiments for Diff-PGD to attack adv-trained ResNet-50 directly. The successful rate is reported below:
>
> | Method   | model (adv-trained) |$n$  | $\epsilon (l_{\infty})$ | SR (%) |
> | -------- | ---- | ---- |----------------------- | ---------------------- |
> | PGD      | Salman2020Do_R50 | 10   | 16 | 90.8              |
> | Diff-PGD | Salman2020Do_R50 | 10    | 16 | 95.2            |
> | PGD      | Salman2020Do_R50 | 20 | 16 | 95.2   |
> | Diff-PGD | Salman2020Do_R50 | 20  | 16 | 96.0    |
>
> where `Salman2020Do_R50` is model card of adv-trained ResNet-50 from Robustbench [1].  We can see that the with the same budgets with sufficient steps, Diff-PGD can still generate better samples with higher success rate.
>
>
>
>
>
> > Did the authors experiment with transformer-based models, such as ViT? I would be interested in the attack performance of these models as well.
>
>
>
> This question is also raised by Reviewer  PJE2. We tested Diff-PGD on popular vit models: ViT-b and BEiT-l. The performance is listed in the first Section of **General Response**.
>
> Also, more **qualitative results** of generated adversarial samples attacking ViTs are put in the **PDF File** of the **General Response** section.

---

### Official Review · Reviewer_B84v · 2023-07-05

**Soundness:** 3 good
**Presentation:** 3 good
**Contribution:** 2 fair
**Rating:** 5
**Confidence:** 4

**Summary:**

This paper proposes to use diffusion models to improve the quality of adversarial examples. It improves the Projected Gradient Descent (PGD) attack by performing a few steps of Stochastic Diffusion Editing (SDEdit) before taking each projected gradient descent (i.e., adversarial) step. The proposed method, DiffPGD, can also be adapted for regional adversarial and style-customized attacks. Experiment results show that DiffPGD can output high-quality images that attack classifiers effectively.

**Strengths:**

1. The idea of using diffusion models to improve the quality of adversarial examples is interesting. The proposed method is shown to generate more natural-looking adversarial examples with better transferability.
2. The evaluation considers different attack scenarios, including digital and physical attacks.


**Weaknesses:**

1. The proposed method is much more computationally expensive (in terms of both time and GPU memory) than regular adversarial attacks, while its advantage over existing methods seems marginal (Figure 6(a)).
2. Many of the experiment results only provide less than five examples, which are inadequate to support the claims (i.e., the four questions listed in Lines 220 - 224) about the proposed methods. For example, the result in Figure 6 (a) is conducted only on 250 samples (rather than the entire 50,000 ImageNet validation samples). The answers to Q2 and Q3 are based only on the examples shown in the figures. It is unclear how well Diff-PGD and its variants perform at a larger scale.
3. The comparison with baselines is not very comprehensive. For example, Figure 4 only shows a qualitative comparison with AdvCam, but no attack success rate is considered in this comparison. While DiffPGD is shown to produce more realistic-looking images in Figure 4, it is not clear if they are as effective as baseline adversarial examples.


**Questions:**

1. The evaluation metric used in Table 2 is unclear. Is it the attack success rate?

**Limitations:**

1. As I mentioned in the weaknesses, the major limitation of this work is its computation overhead. It seems impractical to incorporate this attack method into model training as a defense method, while the original PGD can be easily applied to adversarial training.
2. Limited baselines are considered in the current experiment section, which makes the effectiveness of the proposed method unclear. For example, when demonstrating the transferability and anti-purification of Diff-PGD, this paper only compares to PGD. It is important to compare Diff-PGD with more GAN-based adversarial attack methods to show its advantage.

---

> ### Author Rebuttal · Authors · 2023-08-10
>
> We thank the reviewer for the careful review and valuable comments, we provide answer below.
>
> > The proposed method is much more computationally expensive (in terms of both time and GPU memory) than regular adversarial attacks, while its advantage over existing methods seems marginal (Fig 6(a)).
>
> **First, as said in the General response, we want to emphasize that generating better adversarial attacks, even expensive ones, is an important research direction, as it can help better understand ML systems, and thus help to find potential risks and possible solutions. Works like AutoAttack [2] are known to be computationally expensive (2s/img on GPU) and has never been used in AT, but this attack is often used to evaluate the robustness of ML systems. Evaluating defenses with strong attacks serves as a more reliable evaluation of adversarial robustness.**
>
> Second, it appears that there might be a misunderstanding regarding Fig 6(a): Diff-PGD can achieve 100% success rate as PGD when $n\geq5$, which means it is **still an effective attack** regarding success rate. The advantage over existing methods mainly focus on higher stealthiness and flexibility, which is not marginal.
>
> Regarding the computational cost, we presented a detailed efficiency report of Diff-PGD in General Response, where we show that we can further **optimize** from both code-level and gradient approximation level.
>
> > experiment results only provide less than five examples, which are inadequate to support the claims about the proposed methods. For example, the result in Fig 6 (a) is conducted only on 250 samples (rather than the 50k ImageNet validation). The answers to Q2 and Q3 are based only on the examples shown in the figures. It is unclear how well Diff-PGD and its variants perform at a larger scale.
>
> For Fig 6(a) we want to prove that Diff-PGD is an effective attack, and we uniformly sampled 250 images from ImageNet validation. Further, we scale the size of sampled subset up to 2.5k and run our Diff-PGD with the same settings with $n=5,10$:
>
> | Method   | $n$  | $\epsilon (l_{\infty})$ | SR (%) in 2.5k Samples |
> | -------- | ---- | ----------------------- | ---------------------- |
> | PGD      | 5    | 16                      | 100                    |
> | Diff-PGD | 5    | 16                      | 100                    |
> | PGD      | 10   | 16                      | 100                    |
> | Diff-PGD | 10   | 16                      | 100                    |
>
> from this we can see Diff-PGD can achieve steadily effective attacks.
> Also, for qualitative results corresponding to **style-based attacks**, we provide more samples in the **PDF**.
>
> > The comparison with baselines is not very comprehensive. Figure 4 only shows a qualitative comparison with AdvCam, but no attack success rate is considered. While DiffPGD is shown to produce more realistic-looking images in Figure 4, it is not clear if they are as effective as baseline adversarial examples.
>
> Here we clarify the success rate of style-based attacks: Quantitatively, we initially collected a dataset comprising ~15 scenarios, each consisting of a (source image, style-reference) pair. The final SR(success rate) is:
>
> | Method   | SR(%) |
> | -------- | ----- |
> | AdvCAM   | 100   |
> | Diff-PGD | 100   |
>
> from which we can see that, to successfully attack a DNN with style reference is not a difficult task (the attack is unrestricted), but the qualitative results of generated samples differs, where our Diff-PGD is much better than AdvCAM (Fig 4).
>
> > The evaluation metric used in Table 2 is unclear. Is it the attack success rate?
>
> The metric employed in Table 2 is the attack success rate, and we will clarify this in the revision.
>
> > As I mentioned in the weaknesses, the major limitation of this work is its computation overhead. It seems impractical to incorporate this attack method into model training as a defense method, while the original PGD can be easily applied to adversarial training.
>
> Our work focuses on generating adversarial samples, which is itself an important problem since better adversarial samples can help us reveal the true impression of robustness [6]. **We would like to emphasize that not every defense relies on AT and devising strong attacks can help better evaluate defense strategies.**
>
> However, applying AT to defend against this attack is an interesting question to investigate. One possible strategy would be to generate adv-samples with a stochastic process ($x_K$ is sampled with $x$ plus Gaussian noise), then we can run inference to generate(sample) more samples before a run gradient update, which will be a more efficient data augmentation approach.
>
> > Limited baselines are considered in the experiment section, which makes the effectiveness of the proposed method unclear. For example, when demonstrating the transferability and anti-purification of Diff-PGD, this paper only compares to PGD. It is important to compare Diff-PGD with more GAN-based adversarial attack methods to show its advantage.
>
> For digital attacks, [1, 2] either need to retrain the generative network or need a regularization term, in contrast to our training-free method. Moreover, the generated images using these methods also exist perceptible artifacts [2].
>
> For physical world attacks, most GAN-based baselines [3, 4, 5] are hard to apply to attacks with reference images. Most of them sample new adversarial samples from the noise, which may differ a lot from the given samples. Thus we did not take them into account in our experiments.
>
> [1] Generating Adversarial Examples with Adversarial Networks
> [2] AI-GAN: Attack-Inspired Generation of Adversarial Examples
> [3] Naturalistic physical adversarial patch for object detectors
> [4] Advart: Adversarial art for camouflaged object detection attacks
> [5] Patch of invisibility: Naturalistic black-box adversarial attacks on object detectors
> [6] Reliable evaluation of adversarial robustness with an ensemble of diverse parameter-free attacks

---

> > ### Comment · Reviewer_B84v · 2023-08-11
> > **Thank you for the reply. The rebuttal partially answered my questions.**
> >
> > Thank the authors for their reply and additional results. I have carefully read the author's responses and the comments from other reviewers. The rebuttal answered some of my questions.
> >
> > 1. About my first comment on the marginal improvement over PGD attack: I am aware that both methods achieve 100% attack success rates when n>5, but I was referring to the low-iteration regime, where Diff-PGD performs less effectively. Requiring more iteration makes the method even less efficient. Nonetheless, I agree with the author’s argument that an attack can be solely used for evaluating defense strategies and doesn’t have to be efficient.
> >
> > 2. The second aspect of my concern is the lack of scales in evaluations. I thank the author for conducting additional experiments on 2.5K images. With the new results on 2.5K images, I now tend to believe that the SR would still be 100% on the full set.
> >
> > 3. The quantitative comparison with AdvCAM partially answered my question, but I find the scale of this comparison to be very small (only on 15 pairs of images, if I understand correctly), so the conclusion doesn't seem to be well founded. In the AdvCAM paper, they seem to report their quantitative results on 2000 ImageNet images. I would expect a meaningful quantitative comparison to be conducted on a similar scale.
> > And a follow-up question about the qualitative results in Fig.4: I wonder why your AdvCAM results (the lamp in the bottom row) look much worse than those in Fig.5 of the original paper [1].
> >
> > 4. About my last comment on the lack of comparison with GAN-based methods. It is important to expand the discussion on related work and why they are not considered in this paper. Although the authors mentioned some drawbacks of existing methods in the response, I don’t think they are solid reasons for not conducting the comparisons (especially for digital attacks). On the contrary, it would further support the proposed method by showing its advantage over the GAN-based methods. Therefore, my concern regarding this aspect remains.
> >
> > Thanks again for the efforts in rebuttal. Please see my comments and follow-up questions in the third point. I'd be willing to raise my score if the authors can help address my concerns and questions.
> >
> > [1]. Duan et al. “Adversarial Camouflage: Hiding Physical-World Attacks with Natural Styles” CVPR 2020.

---

> > > ### Author Response · Authors · 2023-08-13
> > > **Thanks for your reply. We hope we can further address your concern here.**
> > >
> > > Thanks a lot for your valuable reply, we are glad to answer the questions you further mentioned here, we hope it will further address your concern:
> > >
> > > > Q1:
> > >
> > > Yes, Diff-PGD need more steps within the same budget to success, since most PGD attacks perform with famous settings like PGD-10 and PGD-20, we think that Diff-PGD is still an effective attack.
> > >
> > > > Q2:
> > >
> > > Thanks for your reply. We are happy to see that we address your concerns on this.
> > >
> > > > Q3: quantitative comparison with AdvCAM
> > >
> > > First, in the AdvCAM paper, they report the results (Section 4.2.3 of [1])  on 2000 samples ImageNet dataset as an ablation study of hyper-parameter $\lambda$. They first sample 2000 samples in ImageNet, define the size of the region to attack, and then choose images from the same category as a style reference. The drawback is that the attacked regions are defined simply as squares or circles. In the experiment part (Section 4.3) of [1], they did the style-based attack on around 100 selected samples.
> > >
> > > To further clarify it, we further construct a better style-based attack dataset based on ImageNet containing 200 cases with 4 categories (sport_car, street_sign, daisy, and guitar), each containing 50 images. We generate the mask using Grounded Segment Anything (Grounded-SAM) [9]  using the category text of ImageNet as a prompt, which can effectively segment out the object to be attacked. Following AdvCAM, we also use randomly sampled images of the same category as a style reference. The quantitative results are as follows:
> > >
> > > |Method|sport_car|daisy|street_sign|guitar|
> > > |---| -- | -- | -- | -- |
> > > |Diff-PGD|98|100|100|98|
> > > |AdvCAM|96|98|94|96|
> > >
> > > Moreover, we find that the samples generated by Diff-PGD consistently exhibit a high level of quality. To provide a quantitative comparison, we plan to put figures in the final versions of our appendix.
> > >
> > > Finally, we hope it can address the concern about the scale of quantitative results.
> > >
> > > > Q3: compared with raw AdvCAM
> > >
> > > For the results of the lamp shown in Figure 5 of [1], the $\lambda$ was set to 2000 as it was mentioned. However, $\lambda=2000$ cannot always guarantee a high success rate (demonstrated in Figure 6 of [1]). In Section 4.2.1 of [1], it is suggested that we should sample from [1000, 10000] for better $\lambda$. With larger $\lambda$, the generated sample will have larger artifacts (e.g. noisy edges and a stippled background). During our experimental investigations, we adopted a value of $\lambda=4000$ (also set as default in their official GitHub repo) to reliably achieve a high success rate. This decision contributes to the discrepancy between our AdvCAM results and those featured in Figure 5 of the original publication [1].
> > >
> > > Furthermore, these observations underscore the necessity for careful hyper-parameter selection in the case of AdvCAM, which may vary across cases. This inherent variability reduces controllability and adaptability compared to Diff-PGD in  style-based attacks.
> > >
> > > > Q4:
> > >
> > > Thanks for your reply.
> > >
> > > In the context of digital attacks, GAN-based techniques diverge significantly from conventional approaches, which is why they are typically excluded from the baseline [10, 11]. The rationale behind this exclusion is outlined below: Firstly, a majority of GAN-based methods center around the concept of adversarial training, involving a generator and a discriminator [2, 3, 4]. Importantly, these methods are not **training-free**; they necessitate training over the dataset. Moreover, GAN-based approaches do not notably excel in terms of stealthiness, as evidenced by findings in [2, 3, 4]. Also, there are other studies, such as [5], that employ GANs for producing **unrestricted attacks**; however, these fall beyond the scope of our task.
> > >
> > > For physical-world attacks where GAN-based achieved great success. especially imethods such as [6, 7] optimize from random noise to sample from GAN as adversarial pattern, while our pipeline focuses more on cases with image reference, so we only consider methods with a reference image as input.
> > >
> > > Still, we thank the reviewer for this question, we will better clarify why GAN-based methods are not included as baselines in our paper.
> > >
> > > [1] Adversarial Camouflage: Hiding Physical-World Attacks with Natural Styles
> > >
> > > [2] AI-GAN: Attack-Inspired Generation of Adversarial Examples
> > >
> > > [3] Generating Adversarial Examples with Adversarial Networks
> > >
> > > [4] A two-stage generative adversarial networks with semantic content constraints for adversarial example generation
> > >
> > > [5] Constructing Unrestricted Adversarial Examples with Generative Models
> > >
> > > [6] Naturalistic physical adversarial patch for object detectors.
> > >
> > > [7] Advart: Adversarial art for camouflaged object detection attacks
> > >
> > > [8] Defense-GAN: Protecting Classifiers Against Adversarial Attacks Using Generative Models
> > >
> > > [9] Segment Anything
> > >
> > > [10] Diffusion Models for Imperceptible and Transferable Adversarial Attack
> > >
> > > [11] Content-based Unrestricted Adversarial Attack

---

> > > > ### Comment · Reviewer_B84v · 2023-08-14
> > > >
> > > > Thanks for the clarification. I find the additional larger-scale experiments to better support the proposed method, and I recommend authors report the updated results in the next version of the paper. I'm raising my score accordingly.

---

> > > > > ### Author Response · Authors · 2023-08-14
> > > > > **Thanks again for your valuable feedback**
> > > > >
> > > > > Thank again you for the valuable feedback. We will try to add the related discussions and analysis in our revision, and please let us know if you have further suggestions!

---

### Official Review · Reviewer_x4rZ · 2023-07-06

**Soundness:** 3 good
**Presentation:** 3 good
**Contribution:** 3 good
**Rating:** 5
**Confidence:** 4

**Summary:**

This paper propose a gradient-based adversarial attack method that use diffusion model to generate adversarial examples.The proposed method can generate adversarial examples with high stealthiness and controllability.

**Strengths:**

1.The proposed method can improve the stealthiness, anti-purification ability and transferability together. In addition, the proposed method can also be implemented in the physical world.

2.The experiments show that the proposed method is effective.

**Weaknesses:**

1.The novelty of the paper is somewhat limited. The gradient-based adversarial attack method has already been proposed [1]. However, the authors have not cited the paper.

2.In the area of adversarial attacks, it is essential to improve the transferability of the adversarial attack. Although the proposed method can enhance the transferability, the authors has never explained the reason of that.

[1] DiffProtect: Generate Adversarial Examples with Diffusion Models for Facial Privacy Protection

**Questions:**

Does the diffusion model necessary in the pipeline?
If we use other adversarial purification method in the pipeline, will the results will be better?

**Limitations:**

Lack the experiment result on the adversarial robust models.

---

> ### Author Rebuttal · Authors · 2023-08-10
>
> We express our gratitude to the reviewer for conducting a careful review and providing valuable feedback. We highly appreciate the reviewer's recognition of the effectiveness of our method. In this response, we aim to address the questions raised by the reviewer:
>
> >The novelty of the paper is somewhat limited. The gradient-based adversarial attack method has already been proposed [1]. However, the authors have not cited the paper.
>
> We read the paper [1] and agree that it is related to our work. We will cite this work in our paper. However, after careful review, we have found the paper differs from our Diff-PGD: first they specific their task on facial protection, also they optimize the latent vector in the latent DM which needs additional regularizations like facial segmentation mask.
>
> **Furthermore, the paper [1] was published (2023/05/23) after the NeurIPS submission deadline (2023/05/17). It would have been impossible for us to become aware of this paper before the deadline.**
>
> > In the area of adversarial attacks, it is essential to improve the transferability of the adversarial attack. Although the proposed method can enhance the transferability, the authors has never explained the reason of that.
>
> We have briefly explained the reason why Diff-PGD can enhance the transferability in Line 285-286 in the manuscript. According to previous works, adv-attacks in semantic space can be better transferred [1, 2]. People define the semantic space using some high-level subspace (e.g. color, blur, shading). In this work, we have diffusion model to curve a better semantic space (the distribution of natural images), which can intuitively explain why Diff-PGD can improve the transferability.
>
> > Does the diffusion model necessary in the pipeline? If we use other adversarial purification method in the pipeline, will the results will be better?
>
> Thank you for raising this interesting question.
>
> It is more convenient to use pre-trained DM in the pipeline of adv-sample generation. First, Diff-PGD serves as a plug-and-play framework, and we only need to plug in a pre-trained SDEdit (input-output-module) to help generate more stealthy adv-samples.
>
> Furthermore, leveraging DM as a robust prior knowledge source significantly aids in the purification process [3]. In contrast, alternative adversarial purification techniques, such as using GAN, would demand re-training [4] or applying regularization [5]. Even with these efforts, the generated samples still tend to retain prominent artifacts [5]. DM, on the other hand, reduces these limitations, facilitating the process and enhancing the quality of the generated samples.
>
> [1] Improving the Transferability of Adversarial Samples with Adversarial Transformations, CVPR 2021
> [2] Natural Color Fool: Towards Boosting Black-box Unrestricted Attacks, NeurIPS 2022
> [3] Diffusion models for adversarial purification, ICML 2022
> [4] Generating Adversarial Examples with Adversarial Networks, IJCAI 2018.
> [5] AI-GAN: Attack-Inspired Generation of Adversarial Examples

---

### Official Review · Reviewer_PJE2 · 2023-07-07

**Soundness:** 3 good
**Presentation:** 3 good
**Contribution:** 3 good
**Rating:** 6
**Confidence:** 3

**Summary:**

This paper proposes a novel Diffusion-Based Projected Gradient Descent (Diff-PGD) framework which optimizes perturbations by using an off-the-shelf diffusion model. The proposed method works well on both digital world and physical world with high stealthiness.

**Strengths:**

1. The proposed method is novel and performs well on both digital world and physical world.
2. The method analysis is clear and visualization is helpful.
3. The experiments are solid and comprehensive.

**Weaknesses:**

1. Lack of the efficiency report of the proposed method
2. Lack of the performance of proposed method under different L-p constraint.
3. Lack of the performance on transformer based vision models.

**Questions:**

1. I am curious the computation cost of the proposed method.
2. How is the performance of the proposed method on transformer based vision models and under different L-p constraint?

**Limitations:**

1. Test the computation time or memory cost of the proposed method.
2. Test the proposed method under different L-p constraint.
3. Test the proposed method on vision transformers.

---

> ### Author Rebuttal · Authors · 2023-08-10
>
> We thank the reviewer for the careful review and valuable comments. Please see our response below.
>
> > Lack of the efficiency report of the proposed method
>
> We present the detailed efficiency (speed and memory occupation) of Diff-PGD  in the section `The computational cost of Diff-PGD` in **General Response**, where we also show that we can further optimize it from both code-level and gradient approximation.
>
> >  Lack of the performance of proposed method under different L-p constraint.
>
> We test L-2 normed attacks on 250 samples in ImageNet validation set with $\epsilon=16*255$ and $n=10$. We present the success rate of Diff-PGD under L-2 norm, and show generated samples in the **PDF file** of **General Response**.
>
> | Method   | $n$  | $\epsilon$ | SR (%) |
> | -------- | ---- | ---------- | ------ |
> | PGD      | 10   | 16*255      | 100    |
> | Diff-PGD | 10   | 16*255      | 100    |
>
> > Lack of the performance on transformer based vision models
>
> Thanks for pointing it out. This is also mentioned by` Reviewer PvL8`. We tested Diff-PGD on two popular vit models: ViT-b and BEiT-l. The performance is listed in Section `Can Diff-PGD attack vision transform-based models ` of **General Response**, and the qualitative results of generated samples are put in the **PDF file** of **General Response**.

---

> > ### Comment · Reviewer_PJE2 · 2023-08-16
> >
> > The anthor answers most of my questions well and I will keep my rating.

---

### Author Rebuttal · Authors · 2023-08-10

# General Response

We thank the reviewers for their valuable comments on our paper. We are excited to see that all the reviewers (Reviewer PJE2, B84v, x4rZ, PvL7) identified the novelty of our technical contribution: utilizing DM to generate adversarial samples, appreciated the application to both digital attacks and physical attacks (Reviewer PJE2, B84v, x4rZ), and found the paper is well-presented (Reviewer PvL7).

Below we address some **commonly asked questions**:



>  ###  Can Diff-PGD work under other l-p perturbations, can it attack vision transform-based models? (Reviewer PJE2, PvL8)

We test L-2 normed attacks on 250 samples in ImageNet validation set with $\epsilon=16*255$ and $n=10$, and find that Diff-PGD can still generate adv-samples with improved stealthiness (refer to the pdf file attached) with high success rate:

| Method   | $n$  | $\epsilon$, $l_2$ | SR (%) |
| ------- | ---- | --------- | ------ |
| PGD      | 10   | 16*255 |100|
| Diff-PGD | 10| 16*255 |100|

For ViT-based model, we replace the ResNet-based models with two popular ViT models (raw ViT-b and BEiT-l), and test the same adv-attack settings as ResNet models in the paper. We can see that Diff-PGD is still effective in attacking ViT-based models:

When set $n=10, \epsilon=16$ (same as our experiments in Section 5 Q(1), DDIM50 with $K_s=3$), the success rate (SR) is still 100%. More detailed results are shown in the following table:

| Model  |  $n$  | $\epsilon$ | SR (%) |
|---|---|---|---|
| ViT_b  |10| 8 | 100 |
| ViT_b  |10 |16 | 100 |
| BEiT_l  |10| 8 | 99.6 |
| BEiT_l  |10| 16 | 99.6|

more **qualitative results** of generated adversarial samples attacking $l_2$ or ViTs are put in the **PDF File** of the **General Response** section.

>  ### The computational cost of Diff-PGD (Reviewer PJE2, B84v)

Firstly, we want to emphasize that generating better adversarial samples itself is important for us to revisit the robustness of our AI systems, helping us figure our potential risks and seek possible solutions. Works like AutoAttack [2] are good examples, it is also slow to run (2s for one image)  and  can not be used in adversarial training though, they still serve as a more reliable evaluation of adversarial robustness.

Here we show the computational cost of our raw Diff-PGD on **single A6000 card**, with different SDEdit step $K$ and iteration $n$:

| $K$  |  $n$  | VRAM (G) |  Speed (s/sample)|
|---|---|---|---|
| 2  |5|  ~18  | ~4 |
| 3 |  5 |~20 | ~5|
| 2  |  10  | ~18  | ~8|
| 3 |  10 |   ~20 | ~10|

The main bottleneck comes from calculating the gradient of SDEdit purified sample $x_0$ over $x$, where the gradient of the large U-Net of DM needs to be saved. Next we present two possible ways to mitigate this issue.

(1) ***Code-level Optimization***: frequently release the GPU memory (**lower VRAM**):

when calculating the gradient of the chain in SDEdit: $\frac{\partial x_0}{\partial x} = \frac{\partial x_K}{\partial x}(\frac{\partial x_{K-1}}{\partial x_{K}}\frac{\partial x_{K-2}}{\partial x_{K-1}}...\frac{\partial x_0}{\partial x_1})$, we only care about the gradient over $x$. Thus we do not need to keep the gradient of all the model parameters along all the chain. What can be optimized from the code level is: first calculate $\frac{\partial x_{K-1}}{\partial x_{K}}$, then calculate $x_{K-2}$, save only the gradient and release the GPU VRAM, and then we can calculate $\frac{\partial x_{K-2}}{\partial x_{K-1}}$. By frequently releasing the GPU memory, we avoid saving gradient over a long chain of computational graph:

| $K$  |  $n$  | VRAM (G) |  Speed (s/sample)|
|---|---|---|---|
| 2  |  5  | ~10  | ~8|
| 3 |  5 |   ~10 | ~10 |

finally, it sacrifices the speed to make it able to run on devices with less GPU memory.


(2) ***Gradient Approximation*** (**lower VRAM, faster speed**)

we can also approximate this complex gradient with a constant: $\frac{\partial x_0}{\partial x} = \frac{\partial x_K}{\partial x}(\frac{\partial x_{K-1}}{\partial x_{K}}\frac{\partial x_{K-2}}{\partial x_{K-1}}...\frac{\partial x_1}{\partial x_0}) \approx c$. The final adversarial update in the $i$-the step becomes (this approximation is also used in SDS (score distillation sampling [1]):

$
\frac{\partial L(f(x^i_0))}{\partial x} \approx  c\frac{\partial L(f(x^i_0))}{\partial x^n_0}
$

from which it can be seen that, we only need to calculate the gradient over $x^n_0$! We only run inference of SDEdit to get $x^n_0$ without gradient saving (lower gpu memory, faster), making it much cheaper. The final cost is:

| $K$  |  $n$  | VRAM (G) |  Speed (s/sample)|
|---|---|---|---|
| 2  |  10  | ~4  | ~4  |
| 3 |  10 |   ~4 | ~5  |



we also add some generated samples with this approximation (in ***additional PDF***) and report success rate of it below:

| Model  |  $n$  | $\epsilon$ | SR (%) |
|---|---|---|---|
| ResNet50  |  10  | 16  | 99.6 |
| ResNet50  |  10  | 8  | 98.8 |
| ResNet50  |  15 | 16  | 100 |
| ResNet50  |  15  | 8  | 100 |

for the global attack tasks, the approximated gradient still shows a high success rate but saves **50%** time and **75%** VRAM !







> ###  Can Diff-PGD work in a large scale with stable efficiency? (Reviewer B84v)

To show that Diff-PGD can be effective with low instability for **global attack**, we rescale the sample number to 2.5k and generate samples with the same settings of Figure 6(a) (ResNet50) with $n=10$, the success rate is:

| Method  |  $n$  | $\epsilon$ | SR (%) |
|---|---|---|---|
| PGD  | 5 |16 | 100 |
| Diff-PGD  | 5  | 16 | 100  |
| PGD  | 10 | 16 | 100 |
| Diff-PGD  | 10 |16 | 100  |

which means that Diff-PGD can steadily attack almost all the cases in the validation dataset.
Also, we collect more cases of **style-based** attacks and show the results in the ***additional PDF*** of the general response.


[1] DreamFusion: Text-to-3D using 2D Diffusion

[2] Reliable Evaluation of Adversarial Robustness with an Ensemble of Diverse Parameter-free Attacks

---

### Decision · Program_Chairs · 2023-09-21

**Decision:**

Accept (poster)

**Comment:**

The recommendation is based on the reviewers' comments, the area chair's personal evaluation, and the post-rebuttal discussion.

This paper studies the use of diffusion models with projected gradient descent for generating realistic adversarial samples. All reviewers find the studied setting novel and the results provide new insights. The authors’ rebuttal has successfully addressed the major concerns of reviewers. Therefore, I recommend acceptance of this submission. I also expect the authors to include the new results and discussion during the rebuttal phase to the final version.